# Continuous Chain of Thought Enables Parallel Exploration and Reasoning

**Halil Alperen Gozeten**[*μ]     **M. Emrullah Ildiz**[*μ]     **Xuechen Zhang**[μ]

**Hrayr Harutyunyan**[γ]     **Ankit Singh Rawat**[γ]     **Samet Oymak**[μγ]

$\mu$: University of Michigan - Ann Arbor
{alperen,eildiz,zxuechen,oymak}@umich.edu

$\gamma$: Google Research
{hrayrh,ankitsrawat}@google.com

## Abstract

Modern language models generate chain-of-thought traces by autoregressively sampling tokens from a finite vocabulary. While this discrete sampling has achieved remarkable success, conducting chain-of-thought with *continuously-valued* tokens (**CoT2**) offers a richer and more expressive alternative. Our work provides new theoretical guarantees and algorithms for CoT2, motivated by logical reasoning tasks that inherently require search capabilities. Theoretically, we establish how CoT2 facilitates the model to track multiple discrete traces in parallel; and quantify the level of achievable parallelism and its benefits for inference efficiency. We also provide a CoT2-based one-layer transformer construction that solves the combinatorial "subset sum problem" given a sufficient embedding dimension. These insights arise from a novel and effective supervision strategy where we match the language model outputs to the empirical token distributions of a set of target traces. Complementing this, we introduce sampling strategies that unlock policy optimization methods for CoT2. Our primary strategy samples and composes $K$ discrete tokens at each decoding step to control the level of parallelism. Experiments confirm that (i) the optimal level of parallelism is governed by the embedding dimension, (ii) our continuous supervision strategy can outperform alternative methods, and (iii) policy optimization with CoT2 indeed improves the performance of the model beyond its initial discrete or continuous supervision.

## 1 Introduction

Chain-of-thought (CoT) strategies (Wei et al., 2022), when paired with strong base models, have achieved immense success and facilitated progress in remarkably challenging tasks, such as solving AIME or IOI problems (Guo et al., 2025; Jaech et al., 2024). Despite these advances, modern language model architectures may fail to utilize their full potential for a few reasons. First is their discrete sampling of tokens—selecting a single token at each decoding step from a vocabulary of $v$ tokens. This limits the model to emitting at most $\log_2(v)$ bits per sample, or more specifically, the Shannon entropy of the softmax output. This contrasts with the $O(d)$ bits each token embedding can store, where $d$ is the embedding dimension. Secondly, discrete sampling can cause the model to *commit* to certain solutions and avoid exploring alternatives (Yao et al., 2023). A practical method to address this is sampling multiple CoT traces and aggregating them, either through consistency (Wang et al., 2022) or best-of-N decoding (Ouyang et al., 2022) through more test-time computation.

In this work, we propose and investigate *CoT with Continuous Tokens* (CoT2) to address these challenges, building on COCONUT (Hao et al., 2024). The fundamental idea in our CoT2 proposal is that rather than the model sampling a single token from the vocabulary, it samples or deterministically

---

[*]Equal contribution.
[†]Code: `https://github.com/alperengozeten/CoT2`

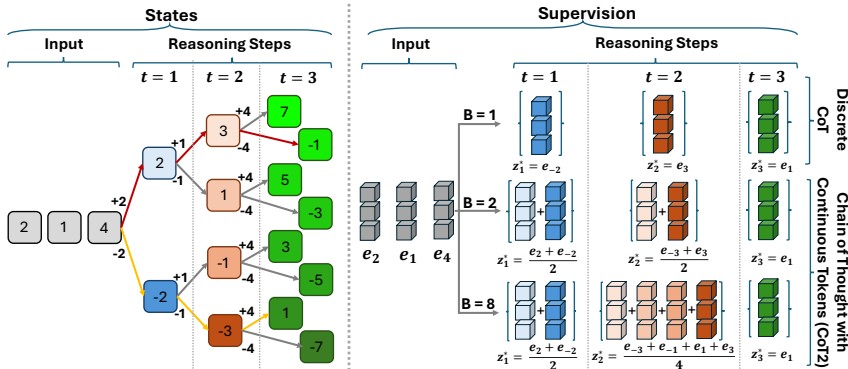

Figure 1: Illustration of CoT2 with varying budgets $B$ for Minimum Non-Negative Sum (MNNS) task with $m = 3$ and input numbers $2, 1, 4$. CoT2 supervision with budget $B$ at steps $t \in \{1, \ldots, m - 1\}$ is the average of embeddings of states visited by $B$ selected trajectories among the 8 possible, and for $t = m$ is the embedding corresponding to the single correct final answer, here $e_1$ for the minimal non-negative sum 1. For $B = 1$ (discrete CoT), the correct trajectory $(-2, -3, 1)$ highlighted with yellow is used; for $B = 2$, the red and yellow trajectories are used; for $B = 8$, all trajectories are included in supervision.

selects a continuous superposition of tokens according to the softmax output. Intuitively, this capability—effectively selecting multiple tokens simultaneously through a continuous superposition—would allow the model to pack more information within each token embedding and also enable it to track multiple reasoning paths in parallel—potentially emulating self-consistency or best-of-N decoding with a single trace. Toward this vision, we make the following technical contributions:

- **Budget-constrained supervision:** We introduce the continuous supervision strategy (CSFT) for CoT2 models to explicitly track multiple teacher traces in parallel, constrained by a budget. This is done by fitting the model to the empirical distribution of the tokens within the expert traces as visualized in Figure 1. Through budget choice, CoT2 can interpolate from discrete CoT to track all reasoning traces. Our method reveals fundamental trade-offs between the budget and embedding dimension in terms of accuracy, validated by an information-packing bound in App. E. Experiments highlight a *sweet spot* for the level of parallelism (see Figure 2) in line with theory.

- **Expressivity and statistical benefits:** We introduce the problem of *Minimum Non-Negative Sum* (MNNS) as a generalization of the classical Subset Sum problem. These problems, as well as related tasks like ProntoQA (Saparov & He, 2022), inherently benefit from parallel search capability. We prove that a single-layer transformer can solve MNNS using CoT2, showcasing the capability of the transformer block to track and expand multiple reasoning traces in latent space.

  On the statistical side, we study CoT2 decoding methods: **(i) Base CoT2:** deterministic inference which creates and feeds continuous tokens using raw softmax output at each step (Sec. 2&3); and **CoT2-MTS (multi-token sampling):** budget-constrained method which samples and averages $K$ discrete tokens to form a continuous token (Sec. 5); and standard CoT, which is a special case of MTS with $K = 1$. Under suitable conditions, we first prove that base CoT2 tracks the "*ideal state*" by aggregating all reasoning paths, whereas MTS provides an unbiased but noisy estimate of this "*ideal state*". Finally, Prop. 3 establishes that the MTS estimate is as powerful as aggregating the outputs of $K$ standard CoT trajectories. In plain language, this formalizes that CoT2 with a budget $K$ can be just as expressive as self-consistency prompting with $K$ traces.

- **Reinforcement learning for CoT2:** We introduce policy optimization methods for CoT2 (Sec. 5). Our primary strategy *MTS* samples and composes K discrete tokens at each forward pass to control the level of parallelism. We also introduce a purely continuous sampling scheme over the probability simplex via Dirichlet sampling. Experiments on the MNNS, ProntoQA, and ProsQA tasks show that GRPO-based RL with CoT2 further improves the accuracy over SFT or CSFT (Sec. 5.3). This demonstrates that the RL phase helps the model better prioritize relevant reasoning traces and offers a promising strategy for training CoT2-based language models.

The paper is organized as follows: Section 2 introduces the technical setup, Section 3 presents our continuous supervision strategy and the MNNS, ProntoQA, and ProsQA tasks. Section 4 provides constructive results and sample-complexity guarantees for CoT2. Section 5 describes our sampling strategies and GRPO-based policy optimization methods, and Section 6 concludes with a discussion.

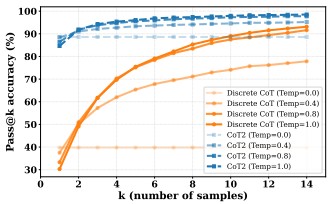

(a) Pass@k accuracies for CoT2 and discrete CoT across different temperatures on the MNNS task.

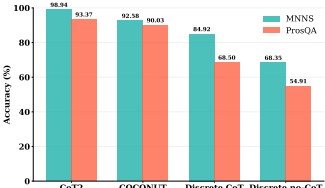

(b) Accuracy of CoT2, CO-CONUT, discrete CoT, and no-CoT on MNNS and ProsQA tasks.

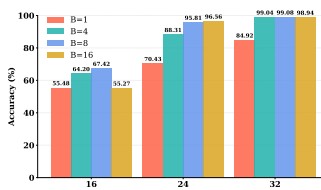

(c) Supervision trajectory budget $B$ vs. different embedding dimensions for CoT2 on the MNNS task.

Figure 2: **(a):** Discrete CoT model requires multiple samplings (Pass@k) to match the single-shot performance of CoT2 model on MNNS (10-run avgs). **(b):** CoT2 model outperforms COCONUT, discrete CoT, and no-CoT in tasks involving search, like MNNS and ProsQA (5-run avgs). **(c):** Tradeoff between the number of trajectories superposed and the embedding dimension (5-run avgs). **Setting:** MNNS with 4 input digits in 1–9. In (a-b), $B$ is the full budget for CoT2, and $B = 1$ for discrete CoT. **(a):** 1-layer, 1-head GPT2 with $d = 24$. For CoT2, the intermediate continuous steps are deterministic, so sampling is performed only for the final discrete answer token. **(b):** MNNS: 2-layer 2-head GPT2, $d = 32$; ProsQA: 4-layer 4-head GPT2, $d = 32$. **(c):** 2-layer, 2-head GPT2 with $d \in \{16, 24, 32\}$.

## 2 PROBLEM SETUP

**Notation.** For an integer $n \geq 1$, we use the shorthand $[n] = \{1, \ldots, n\}$, and denote vectors by bold lowercase letters (e.g. $\boldsymbol{x}$) and matrices by bold uppercase letters (e.g. $\boldsymbol{X}$). For a vector $\boldsymbol{x} \in \mathbb{R}^n$, the component $x_i$ is its $i$-th entry. The zero vector in $\mathbb{R}^n$ is $\boldsymbol{0}_n$, and the zero matrix in $\mathbb{R}^{m \times n}$ is $\boldsymbol{0}_{m \times n}$. Finally, we let $\Delta^{v-1}$ denote the standard $v - 1$ simplex in $\mathbb{R}^v$.

Assume we have an input context $\boldsymbol{X} \in \mathbb{R}^{n \times d}$, where each row is a $d$-dimensional embedding vector. Our goal is to output $m$ tokens given the context $\boldsymbol{X}$ with $m$th output token being the final answer that is evaluated under a performance metric (e.g. accuracy or reward). For the first $m - 1$ steps, the model outputs *continuous thought tokens* $\{z_t\}_{t \in [m-1]}$ that enable a reasoning process. In the final step $t = m$, the model outputs a *discrete* token $z_m$ from a vocabulary of size $v$. In the remainder of this paper, we investigate strategies for training this system in a way that improves final performance over standard discrete next-token prediction.

Formally, let $\boldsymbol{E} = [\boldsymbol{e}_1, \ldots, \boldsymbol{e}_v]^\top \in \mathbb{R}^{v \times d}$ be the embedding matrix for the vocabulary of $v$ tokens, where $\boldsymbol{e}_i \in \mathbb{R}^d$ represents the embedding of the $i$th token. We define the next-token prediction model $\text{LM}_\theta$ parameterized by $\theta$ that assigns, at each step $t$, a probability distribution over possible next tokens given the prefix $z_{<t}$ and context $\boldsymbol{X}$. Concretely, for $1 \leq t \leq m - 1$, the model outputs the following distribution over the $v$ vocabulary entries via a softmax operation:

$$\text{LM}_\theta(\cdot \mid z_{<t}, \boldsymbol{X}) := \boldsymbol{\alpha}_t \quad \text{where} \quad \boldsymbol{\alpha}_t = \left[ \alpha_{t,1}, \ldots, \alpha_{t,v} \right] \in \Delta^{v-1},$$

i.e. $\alpha_{t,i} \geq 0$ and $\sum_{i=1}^v \alpha_{t,i} = 1$. We then form the continuous token as a convex combination of all tokens in the vocabulary:

$$z_t = \boldsymbol{E}^\top \boldsymbol{\alpha}_t \in \mathbb{R}^d, \quad \forall 1 \leq t \leq m - 1.$$

At the final step $t = m$, the model samples a discrete token $z_m \in \{\boldsymbol{e}_1, \ldots, \boldsymbol{e}_v\}$ from policy distribution $\text{LM}_\theta(\cdot \mid z_{<m}, \boldsymbol{X}) = \boldsymbol{\alpha}_m$. We note that we assume that the answer depends only on the final discrete token $z_m$ merely for simplicity; the same framework naturally extends to decoding multiple final discrete tokens after continuous ones. We refer to this decoding strategy as **base CoT2** and observe that it results in a deterministic reasoning chain because the continuous tokens are precisely determined by the softmax map (see Remark 1 in Appendix C for a brief discussion of its scalability). In Section 5, we will introduce stochastic alternatives, such as **CoT2-MTS**, to facilitate generative reasoning.

## 3 CSFT: A SUPERVISED TRAINING METHOD FOR COT2

In this section, we present our method of continuous supervised training to learn intermediate thought tokens as "soft" targets rather than "hard" target tokens. Specifically, we provide the model with convex combinations of vocabulary embeddings, which allows the model flexibility in those reasoning

steps. Such an approach is particularly suitable when the task accuracy depends only on the final token or token distribution. Formally, at each reasoning step $t = 1, \ldots, m-1$, the supervision specifies a target probability distribution

$$\alpha_t^* = \left[ \alpha_{t,1}^*, \ldots, \alpha_{t,v}^* \right] \in \Delta^{v-1},$$

where $\alpha_{t,i}^* \geq 0$ and $\sum_{i=1}^{v} \alpha_{t,i}^* = 1$. We train the model to align its predicted distribution $\alpha_t$ to the supervision distribution $\alpha_t^*$ rather than one-hot labels by using a divergence-based loss:

$$\mathcal{L}_{\mathrm{cont}}(\theta; X, t) = D\left( \alpha_t^* \,\|\, \alpha_t \right),$$

where $D\left(\cdot\|\cdot\right)$ is the cross-entropy (or equivalently KL divergence) between two distributions. This approach can also be viewed as *token-level knowledge-distillation*, where the teacher distribution $\alpha_t^*$ is obtained through a logic/search algorithm. At the final step $t = m$, we have a discrete target $z_m^* \in \{e_1, \ldots, e_v\}$, so that $\alpha_m^*$ is one-hot distribution placing probability 1 on target token and 0 elsewhere. This is equivalent to using a standard cross-entropy loss $-\log \mathrm{LM}_\theta \left( z_m^* \mid z_{<m}, X \right)$ at the final step. Hence, for each training example, the total loss for continuous supervised training is the sum of the continuous-token divergence losses:

$$\mathcal{L}_{\mathrm{CSFT}}(\theta; X) = \sum_{t=1}^{m} \mathcal{L}_{\mathrm{cont}}(\theta; X, t). \tag{1}$$

By minimizing $\mathcal{L}_{\mathrm{CSFT}}(\theta)$, we teach the model the soft targets $\alpha_t^*$ at each step and to predict the correct final discrete token. Inspired by the discussions in Bachmann & Nagarajan (2024); Bengio et al. (2015), we consider two ways of providing prefixes to the language model:

1. **Teacher forcing:** Each step $t$ is conditioned on the ground-truth prefix $z_{<t}^*$, meaning the model has access to all previous ground-truth tokens during prediction. Formally, for each step $t' < t$, the corresponding input $z_{t'}^* = E^\top \alpha_{t'}^*$ is a convex combination of all vocabulary tokens.
2. **Self-feeding:** Each step $t$ autoregressively uses the model's previously generated outputs, $z_{<t}$, during training. In particular the continuous output token $z_t = E^\top \alpha_t$, is a convex combination of vocabulary embeddings, which is then fed back to the model as part of the prefix.

It is also worth noting that one may apply *temperature scaling* or *thresholding* to $\alpha_t$ before forming $z_t$ in order to filter the model's predictions. In our experiments, we find that teacher forcing leads to superior performance for CSFT, even though at inference time, the model runs in an autoregressive manner, as discussed below. See Appendix D for further discussion.

**Inference.** At inference, the model does not rely on the ground-truth distributions $\alpha_t^*$. Instead, at each continuous step $t < m$, it autoregressively produces the output distribution $\alpha_t$, converts it to a continuous token $z_t = E^\top \alpha_t$, and appends $z_t$ to the prefix for the next prediction. In the final step, the model samples a discrete token from $\alpha_m = \mathrm{LM}_\theta(\cdot \mid z_{<m}, X)$.

**Baselines.** Our first baseline is discrete CoT, which uses teacher-forced training where the next token prediction is performed conditioned on the previous ground-truth tokens with standard cross-entropy loss. Discrete baseline enforces $z_t^*$ to be a token in vocabulary $\{e_1, \ldots, e_v\}$, which means that it is a special case of CSFT where the $\alpha_t^*$ are one-hot vectors rather than an arbitrary element of $\Delta^{v-1}$. The model minimizes the following objective, which is obtained by summing over all steps of teacher-forced next-token prediction:

$$\mathcal{L}_{\mathrm{SFT}}(\theta; X) = \sum_{t=1}^{m} -\log \mathrm{LM}_\theta \left( z_t^* \mid z_{<t}^*, X \right). \tag{2}$$

Here, setting $m = 1$ gives the discrete no-CoT baseline, where the model directly predicts the final answer. Another baseline we compare against is COCONUT, which replaces discrete tokens sequentially with the last hidden state of the LLM following a left-to-right curriculum learning strategy. In inference, the COCONUT model directly outputs the answer at the final step ($t = m$), following $m - 1$ intermediate continuous thought tokens produced by the hidden state output.

### 3.1 TASKS REQUIRING EXPLORATION OVER STATES

In this subsection, we illustrate CSFT training described in (1) on tasks that require *exploration* over multiple states but have a single final correct state. We consider a directed graph exploration problem

where the aim is to follow a $m$-step trajectory through reachable states starting from $g_0$ and arriving at a desired state $g_m^*$. Suppose that the vocabulary is sufficiently large that each state $g$ of the task can be assigned a unique embedding. Let $\Gamma_t$ denote the set of states reachable at step $t$ building upon step $(t-1)$ with $\Gamma_0 = \{g_0\}$, and let $\mathcal{T}$ be the set of complete trajectories $\pi$ of length $m$, each inducing a state sequence $(g_1(\pi), \ldots, g_m(\pi))$ with $g_t(\pi) \in \Gamma_t$. Next, we describe how to obtain CSFT targets $\{\alpha_t^*\}_{t=1}^m$ by curating a set of complete trajectories and projecting them back to intermediate steps.

**CSFT Targets Through Hindsight Top-$B$ Superposition.** Let $F : \mathcal{T} \to \mathbb{R}_{\geq 0}$ be a task-specific final score function with lower is better, and fix a budget $B \geq 1$ for the number of trajectories. We form the set $\Pi_B$ by filtering trajectories using $F$ and taking the top $B$:

$$\Pi_B = \underset{\Pi \subseteq \mathcal{T},\ |\Pi|=B}{\arg\min} \sum_{\pi \in \Pi} F(\pi).$$

At step $t < m$, we form the supervision $\alpha_t^*$ by superposing the states visited by $\Pi_B$ and at step $t = m$, we select one correct final state from $\Gamma_m$, so that $\alpha_m^*$ is a one-hot vector:

$$\alpha_{t,g}^* = \frac{1}{B} \sum_{\pi \in \Pi_B} \mathbf{1}\{g_t(\pi) = g\}, \quad g \in \Gamma_t,\ t < m; \qquad \alpha_{m,g}^* = \begin{cases} 1, & \text{if } g \text{ is the correct final state } g_m^*, \\ 0, & \text{otherwise.} \end{cases} \quad (3)$$

This rule is retrospective in the sense that it ranks trajectories and then supervises the previous steps accordingly. If $B = 1$, the supervision reduces to **discrete CoT** using ground-truth trajectory in (2) by simply choosing $F$ to penalize any trajectory with incorrect final answer. If $B = |\mathcal{T}|$ is the full trajectory budget, then $\Pi_B = \mathcal{T}$, and the supervision $(\alpha_{t,g}^*)$ becomes the superposition of **all reachable states**. As a result, the described methodology provides flexibility to select the number of superposed states based on model capacity and the task structure.

### 3.1.1 Minimum Non-Negative Sum Task

We now introduce the *Minimum Non-Negative Sum* (MNNS) task, where the goal is to assign signs to a list of numbers so that their sum is as small as possible while being nonnegative. The MNNS task can also be viewed as partitioning a set of numbers into two subsets with a minimal difference, which makes it closely related to the subset-sum problems explored in Dziri et al. (2023); Thomm et al. (2024). Formally, given $m$ integers $d_1, \ldots, d_m$, the task is to assign signs $\sigma_i \in \{+1, -1\}$ such that $s = \sigma_1 d_1 + \cdots + \sigma_m d_m \geq 0$ and $s$ is minimized. Let $\sigma^{\text{opt}} = (\sigma_1^{\text{opt}}, \ldots, \sigma_m^{\text{opt}})$ denote the optimal assignment that achieves the minimal nonnegative sum $s^{\text{opt}}$ out of $2^m$ possible sign assignments. Here, every possible *partial sum* $\sigma_1 d_1 + \cdots + \sigma_t d_t \in \Gamma_t$ is treated as a state and assigned a unique embedding $e_{\phi(\sigma_1 d_1 + \cdots + \sigma_t d_t)}$, where $\phi(\cdot)$ maps each sum to a distinct id in $[v]$. At step $t$, the state $\sigma_1 d_1 + \cdots + \sigma_{t-1} d_{t-1} + \sigma_t d_t$ is reachable from the state $\sigma_1 d_1 + \cdots + \sigma_{t-1} d_{t-1}$ at step $t-1$.

We supervise the model with CSFT targets $\alpha_{t,g}^*$ defined in (3) with different budgets $B$ between 1 and $|\mathcal{T}| = 2^m$. For MNNS, we rank trajectories by the absolute value of the final sum for budgets $1 < B \leq 2^m$, while for $B = 1$ we select the ground-truth trajectory. We split the training and validation datasets by ensuring that any permutation of numbers appears in exactly one split in order to prevent memorization and make a fair evaluation. We encode input and output numbers with separate tokens in our vocabulary. As an example, an input appears as $\langle\text{BOS}\rangle\ d_1\ d_2\ \ldots\ \to$, and the corresponding output as $s_1\ s_2\ \ldots\ s^{\text{opt}}\ \langle\text{EOS}\rangle$, where $s^{\text{opt}}$ is the minimal nonnegative sum for $\{d_1, \ldots, d_m\}$. For the model, we use the GPT2 architecture (Radford et al., 2019) with different head, layer, and embedding dimension configurations, and train it from scratch. During evaluations, we only assess the final answers of both approaches. For more details on the experiments, see Appendix C.

### 3.1.2 ProntoQA and ProsQA Datasets

Other datasets we explore in our investigation of the CSFT approach are the ProntoQA (Saparov & He, 2022) and ProsQA (Hao et al., 2024), which are logical reasoning tasks that require exploration over multiple paths. Each question in ProntoQA asks whether a certain target word (node) $T$ is reachable from a root word (node) $R$ within a fixed number of hops, while for ProsQA it asks which of the target words $T_1$ or $T_2$ is reachable. We use 5-hop questions and present the graph in a structured format by representing nodes and edges using embeddings inside the context for each question, and use these as the model input rather than raw text input.

The graph structure of the ProntoQA and ProsQA tasks naturally obeys the supervision in (3). For CoT2 model, we use the full trajectory budget $B = |\mathcal{T}|$, so at step $t$ the target $\alpha_t^*$ is the weighted distribution over all nodes reachable from $R$ in $t$ hops, thus tracking all possible trajectories without filtering. In final step $m$, the supervision assigns probability 1 to the correct label: yes or no for ProntoQA, and $T_1$ or $T_2$ for ProsQA. For discrete CoT model ($B = 1$), we provide the ground-truth path from $A$ to the target node ($T_1$ or $T_2$) as the supervision. Please refer to Appendix C.2 for additional details on supervision and data format.

### 3.2 Results and Discussion of CoT2 Supervision

**CoT2 with full trajectory budget** $B = |\mathcal{T}|$**.** As demonstrated in Figure 1, training with full budget $B = |\mathcal{T}|$ corresponds to forming supervision by superposing all reachable states at each step. In experiments on MNNS, ProsQA, and ProntoQA tasks, we observe that the CoT2 model trained with CSFT using full trajectory budget $B$ significantly outperforms other baselines, as shown in Figures 2b and 3. Moreover, this enables faster convergence as illustrated in Figure 3. In particular, when trained with full trajectory budget $B$, the model does not make intermediate decisions but instead conducts

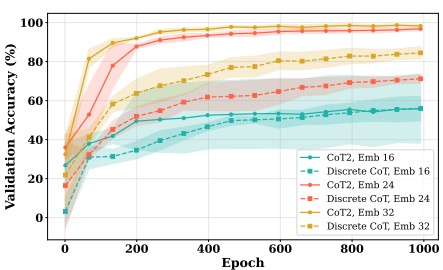

Figure 3: Training performance vs. embedding dimension for CoT2 ($B = 16$) and discrete CoT ($B = 1$) on MNNS with 4 input digits from 1-9 and 2-layer, 2-head GPT2 with $d \in \{16, 24, 32\}$.

search over states through continuous tokens and defers the decision to the last step. This mitigates error accumulation by avoiding early commitments. Supporting this, Figure 2a applies Pass@k to both methods and shows that CoT2 consistently outperforms discrete CoT at every $k$; importantly, the discrete CoT requires multiple attempts to approach the single-attempt ($k = 1$) performance of CoT2 model, aligning with the "snowballing errors" phenomenon observed in discrete autoregressive generation (Bachmann & Nagarajan, 2024). Our results also indicate that when supervision from a search algorithm is available, leveraging this denser signal via CSFT is preferable to approaches that internalize discrete CoT with continuous tokens, such as COCONUT. Furthermore, once the CoT2 model is given a moderate threshold embedding capacity (i.e., $d = 24$ in Figure 3) to represent all possible states at each step, it solves the search-based tasks with near-perfect accuracy and achieves superior performance with fewer layers/heads compared to the discrete CoT model (Appendix D.1). We also provide additional experiments on ProntoQA and ProsQA in Appendix D.1 that confirm similar findings to the MNNS task in Figures 2a and 3.

**Budget–embedding dimension tradeoff.** Fig. 2c varies supervision trajectory budget $B \in \{1, 4, 8, 16\}$ (with $|\mathcal{T}| = 16$) across embeddings $d \in \{16, 24, 32\}$. With a small embedding ($d = 16$), the model capacity is not enough to learn large budget $B = 16$ superposing all reachable states, whereas a moderate budget $B = 8$ performs best by reducing representational load relative to $B = 16$ and avoiding the per-step harder commitments of $B = 1, 4$. Note that $B = 16$ imitates search by representing all possible states with decision taken only in the last step, whereas $B = 4, 8$ forces earlier decisions but reduces the representational load. Accordingly, as $d$ grows (e.g., 24 or 32), the performance improves monotonically with budget $B$ and reaches near-ceiling performance when $B = 16$. This indicates a capacity–parallelism tradeoff in which larger $B$ helps only when $d$ is sufficiently large to represent the superposed traces.

**Theoretical Perspective on Embedding Capacity.** The embedding-capacity threshold behavior in Fig. 2c aligns with the information-packing bound we discuss in Appendix E, showing that robust decoding of a budget-$B$ superposition over $v$ candidate states requires embedding $d = \Omega\left(B \log(v/B)\right)$ in the worst case. This implies, once $d > \log_2 v$, discrete CoT ($B = 1$) becomes information-inefficient as each discrete token carries at most $\log_2 v$ bits, whereas CoT2 can pack $B \approx \Theta\left(d / \log(v/B)\right)$ states per step, which is intuitively consistent with the entropy being $\approx \log_2 \binom{v}{B} \approx B \log_2(v/B) \lesssim d$.

## 4 Theoretical Analysis of CoT2

In this section, we construct one-layer transformer solving the MNNS task using an attention layer followed by an MLP layer, inspired by the capability of one-layer, one-head transformer under CSFT supervision to solve this task with sufficient embedding dimension. We then provide a theoretical comparison on the sample complexities of base CoT2, CoT2-MTS, and discrete CoT models.

### 4.1 Solving the Minimum Non-Negative Sum Task

**Proposition 1** (Solving MNNS). *There exists a $1$-layer transformer architecture with embedding dimension $d = d_e + d_p$, where $d_e = 2^{n+1}$ is the state-encoding dimension and $d_p = n + 2$ is the positional-encoding dimension, that solves the MNNS task using CoT2 by storing sine–cosine embeddings of all $2^k$ states at the $k$-th iteration in a non-overlapping manner.*

The above construction utilizes trigonometric embeddings, inspired by the mechanistic insights given by Nanda et al. (2023). Our approach leverages these trigonometric embeddings to provide a theoretical guarantee that **the transformer can track and add/subtract multiple numbers in parallel** by benefiting from the embedding capacity and reading off the minimum non-negative number at the final step. An important observation regarding our construction is that the trajectories at each intermediate reasoning step are truly decoupled as it stores each state using non-overlapping (sine, cosine) representations. This closely parallels the left side of Figure 1, but we also utilize rotations/shifts to ensure distinct states are orthogonal and are easy to read out. Finally, as empirical validation of this theoretical result, we observe that training according to this construction with trigonometric embeddings yields perfect accuracy.

### 4.2 Understanding and Formalizing the Benefits of CoT2 and Comparison to CoT

In the rest of this section, we argue that CoT2 equips the model with the ability to track multiple paths in parallel, which is formalized through Assumption 1 below. Building on this condition, we will provide a formal comparison of CoT2 and discrete CoT models in the remainder of this section. To improve expressivity, we allow a step-indexed (t) policy $\text{LM}_\theta^{(t)}$ throughout, while remaining consistent with the problem setup.

**Assumption 1.** *Recall the model $\text{LM}_\theta$ in Section 2. For any step $t$ and prefix tokens $z_{\leq t}$, we assume (i) the next token probabilities depend only on the last token $z_t$ and the query $X$ and (ii) if the last token is $z_t = \sum_{j=1}^{v} \alpha_{t,j} e_j$ so that $\sum_{j=1}^{v} \alpha_{t,j} = 1$, the output distribution $\alpha_{t+1}$ decouples as follows:*

$$\text{LM}_\theta^{(t)}(\cdot \mid z_{\leq t}, X) \overset{(i)}{=} \text{LM}_\theta^{(t)}(\cdot \mid z_t, X) \overset{(ii)}{=} \sum_{j=1}^{v} \alpha_{t,j} \text{LM}_\theta^{(t)}(\cdot \mid e_j, X).$$

This assumption holds for inherently serial tasks, such as MNNS construction in Proposition 1, where previous reasoning history is summarized in the final token and different trajectories are decoupled at intermediate reasoning steps. Under Assumption 1, the token distribution $\alpha_{t+1} = \text{LM}_\theta^{(t)}(\cdot \mid z_t, X)$ evolves with the equation $\alpha_{t+1} = \alpha_t M_t(z_t; X)$, starting from $\alpha_1 = \text{LM}_\theta^{(0)}(\cdot \mid X)$ until $\alpha_m$. Here, $M_t(z_t; X) \in \mathbb{R}^{v \times v}$ is a Markov transition matrix that depends on the input $X$ and the last token $z_t$ (see (Ildiz et al., 2024) for related discussion). To keep exposition cleaner, we omit $z_t$ and $X$ in the notation of $M_t(z_t; X)$, and use $M_t$ instead. We start by defining the three inference strategies.

- **Base CoT2**: At each step $t = 1, \ldots, m$, the model outputs the continuous token $z_t = E^\top \alpha_t$ and uses it as the query for the next step.
  **Interpretation:** Base CoT2 simultaneously tracks and aggregates all possible $v^m$ traces over $m$ steps; where the trace $(i_t)_{t=1}^m$ has a weight of $\prod_{i=1}^m \alpha_{t,i_t}$.
- **Discrete CoT**: At each step $1 \leq t \leq m$, the model samples exactly one token $z_t = e_{i_t}$ from $\alpha_t$, and uses it as the query for the next step.
  **Interpretation:** Discrete CoT samples a single trace out of $v^m$ traces with a likelihood of $\prod_{i=1}^m \alpha_{t,i_t}$ for trace $(i_t)_{t=1}^m$.
- **CoT2-MTS (multi-token sampling)**: At each step $1 \leq t \leq m$, i.i.d. sample $K$ tokens $e_{i_1}, \ldots, e_{i_K}$ from $\alpha_t$, average these tokens to form $z_t = \frac{1}{K} \sum_{r=1}^K e_{i_r}$, which it uses as query for the next step.
  **Interpretation:** CoT2-MTS tracks $K$ traces in parallel according to their discrete CoT likelihoods. However, these traces are not statistically independent.

With these methods defined, we now present a result on the statistical consistency of their outputs.

**Proposition 2** (Consistency of CoT and CoT2 inference). *Under Assumption 1 and given $X$, the output of base CoT2 is $z_m = \sum_{j=1}^{v} \alpha_{m,j} e_j$ where $\alpha_m = \alpha_1 \prod_{t=1}^{m-1} M_t$. Discrete CoT and CoT2-MTS have the same output once we take the expectation over their stochastic sampling.*

**Remark:** The above proposition states that as the number of samples approaches infinity, the empirical distribution over the vocabulary $\hat{\alpha}_m$ obtained from CoT2-MTS or discrete CoT traces

converges in probability to $\alpha_m$. Here, $\alpha_m$ is the deterministic output of the base CoT2 model, which is computed without sampling and is not a random variable.

Proposition 2 establishes the statistical consistency of all three methods as they estimate the same distribution $\alpha_m$. However, they differ in the samples needed to approximate this distribution. In particular, the base CoT2 model outputs the entire probability distribution over tokens at every intermediate step, implicitly tracking all possible trajectories in parallel as continuous embeddings. Consequently, it computes the exact final token distribution in one forward pass without repeated sampling. In contrast, due to stochasticity, discrete CoT or CoT2-MTS require multiple i.i.d. samples to approximate this distribution. This motivates us to study and contrast their sample complexities. The next proposition provides a distribution approximation guarantee in $\ell^2$ distance and shows that CoT2-MTS reduces the sample complexity of estimation compared to discrete CoT by a factor of $K$.

**Proposition 3.** *(i) Let $\alpha_m$ be the expected output distribution after $m$ steps of CoT according to Proposition 2. Let $\hat{\alpha}_m^{(\text{MTS})}$ be the distribution resulting from averaging the outputs of $N$ i.i.d. CoT2-MTS traces with parallelism $K$. Then, to guarantee $\|\hat{\alpha}_m^{(\text{MTS})} - \alpha_m\|_2 \leq \epsilon$ with high probability, the total number of samples (traces) required scales as $\Theta(K^{-1}\epsilon^{-2})$. (ii) Defining $\hat{\alpha}_m^{(\text{disc})}$ to be the distribution obtained by averaging the outputs of $NK$ i.i.d. discrete CoT traces, we have the following upper bound on the expected error of MTS:*

$$\mathbb{E}\left[\|\hat{\alpha}_m^{(\text{MTS})} - \alpha_m\|_2^2\right] \leq \left(2 - \frac{1}{K}\right) \mathbb{E}\left[\|\hat{\alpha}_m^{(\text{disc})} - \alpha_m\|_2^2\right].$$

The above proposition implies that one MTS rollout tracks $K$ traces in parallel, behaving in terms of estimation error like $K$ independent discrete CoT traces. Recall that CoT2-MTS generalizes discrete CoT, which corresponds to $K = 1$. For this case, the proposition reduces to the known $\Theta(\epsilon^{-2})$ sample complexity of approximating a $v$-category distribution in $\ell^2$ distance (Kamath et al., 2015). Note that as $K \to \infty$, CoT2-MTS converges to the base CoT2, and the proposition recovers the one-shot performance of base CoT2. Thus, although the three models yield the same final distribution, discrete CoT requires $\Theta(K)$ times more rollouts than CoT2-MTS with parallelism $K$ to achieve a similarly accurate approximation, due to inherent noise from single-token sampling. In contrast, the base CoT2 model carries the entire mixture of partial expansions at each step and computes the distribution in one shot. This theoretical intuition aligns with empirical findings in the Pass@k experiments, where CoT2 achieves comparable performance to discrete CoT with substantially fewer samples.

## 5 Reinforcement Learning Methods for CoT2

In this section, we present two sampling methods (Multi-token and Dirichlet sampling) to apply reinforcement learning (RL) with continuous output tokens. Specifically, we explore Group Relative Policy Optimization (GRPO) training on top of (i) discrete models and (ii) continuous models that are supervised trained following the previous section for the MNNS, ProntoQA, and ProsQA tasks. Concretely, we demonstrate that RL (a) adapts discrete SFT models to produce continuous outputs via MTS/Dirichlet rollouts, and (b) sharpens CoT2 models by prioritizing relevant reasoning traces through reducing entropy of continuous token representations rather than weighting them equally. We note that the existing literature in RL operates in the model's native discrete action space over a finite vocabulary to maximize a scalar reward on the generated sequence (Ouyang et al., 2022; Shao et al., 2024). In contrast, we perform RL in a continuous action space, which is the linear combination of token embeddings, to maximize the same scalar reward.

In our setup, a language model $\text{LM}_\theta$ acts as a *policy* over tokens. Let $\{\mathbf{Z}^{(i)}\}_{i=1}^G$ be a group of $G$ trajectories sampled from old policy $\text{LM}_{\theta_{\text{old}}}$ where each trajectory $\mathbf{Z}^{(i)} = \left(z_1^{(i)}, \ldots, z_m^{(i)}\right)$ contains $m$ output tokens for a fixed input $X$. We assume a sparse reward setting where the reward is 1 for a correct final answer and 0 otherwise. Let $\hat{A}_{i,t}$ denote the advantage estimate at step $t$ in trajectory $i$ and under sparse reward setting, $\hat{A}_{i,t} = \hat{A}_i$ is identical across all steps. To quantify how the new policy $\text{LM}_\theta$ differs from the old one on token $z_t^{(i)}$ in $i$th trajectory, we define the policy ratio $r_t^{(i)}(\theta) = \frac{\text{LM}_\theta\left(z_t^{(i)}|z_{<t}^{(i)}, X\right)}{\text{LM}_{\theta_{\text{old}}}\left(z_t^{(i)}|z_{<t}^{(i)}, X\right)}$.

We update the model by minimizing the objective (Shao et al., 2024; Yu et al., 2025):

$$\mathcal{L}_{\text{GRPO}}(\theta) = -\frac{1}{\sum_{i=1}^G |\mathbf{Z}^{(i)}|} \sum_{i=1}^G \sum_{t=1}^{|\mathbf{Z}^{(i)}|} \left[\min\left(r_t^{(i)}(\theta)\,\hat{A}_{i,t}, \ \text{clip}\left(r_t^{(i)}(\theta), 1-\epsilon, 1+\epsilon\right)\hat{A}_{i,t}\right) - \beta\,\mathbb{D}_{\text{KL}}\left[\text{LM}_\theta \,\|\, \text{LM}_{\theta_{\text{ref}}}\right]\right].$$

As the output length is fixed in our setting, we have $|\mathbf{Z}^{(i)}| = m$ for each trajectory. Here, $\epsilon$ clips the ratio $r_t(\theta)$, and $\beta$ controls the KL-divergence from a SFT-initialized reference policy $\text{LM}_{\theta_{\text{ref}}}$. We set GRPO iterations $\mu = 1$ and estimate KL divergence with Schulman Approximator (Shao et al., 2024).

## 5.1 Multi-Token Sampling

We emulate the rollout of a continuous token by sampling a fixed number of $K$ discrete tokens and averaging them at steps $t = 1, \ldots, m-1$. We refer to this hybrid method as *CoT2-MTS* (multi-token sampling). For the GRPO objective, we propose calculating the policy ratio for continuous tokens as follows. Assume that at step t, the old policy samples $K$ discrete tokens $e_{i_1}, \ldots, e_{i_K}$ from $\alpha_t^{\text{old}}$. Let $\alpha_{t,i_1}, \ldots, \alpha_{t,i_K}$ denote the probabilities assigned to these same sampled tokens by the current policy. We define the policy ratio for continuous steps by dividing geometric means:

$$r_t(\theta) = \frac{\text{LM}_\theta (z_t \mid z_{<t}, X)}{\text{LM}_{\theta_{\text{old}}} (z_t \mid z_{<t}, X)} = \left( \frac{\alpha_{t,i_1} \cdots \alpha_{t,i_K}}{\alpha_{t,i_1}^{\text{old}} \cdots \alpha_{t,i_K}^{\text{old}}} \right)^{1/K}, \quad (4)$$

|  |  | ProsQA | | ProntoQA | |
|---|---|---|---|---|---|
|  |  | SFT | SFT+GRPO | SFT | SFT+GRPO |
| K=6 | CoT2 | 93.37 | 93.83 | 75.36 | 76.15 |
|  | Discrete CoT | 68.50 | 68.24 | 59.58 | 62.28 |
| K=8 | CoT2 | 93.37 | 94.09 | 75.36 | 76.66 |
|  | Discrete CoT | 68.50 | 71.58 | 59.58 | 71.53 |
| K=12 | CoT2 | 93.37 | 94.21 | 75.36 | 77.64 |
|  | Discrete CoT | 68.50 | 72.76 | 59.58 | 74.03 |

Table 1: Validation accuracies on ProsQA and ProntoQA for CoT2 and Discrete CoT, evaluated at $K = 6, 8, 12$ with CoT2-MTS sampling GRPO. All models use a 4-layer, 4-head GPT2 with embedding dimension 32. Remarkably, GRPO with MTS sampling scheme results in consistent improvements.

for $t = 1, \ldots, m-1$. The geometric mean ensures the ratio for each continuous step remains on the same scale as the final discrete token's ratio and, thus, helps avoid overly large or small updates and stabilizes GRPO training compared to the direct multiplication of probabilities. Once this ratio is computed, we average the $K$ sampled tokens to form $z_t$, which is fed to the model as the query for the next prediction step. At the final step $t = m$, where the token $z_m = e_j$ is discrete with $j \in [v]$ denoting its index, the policy ratio is simply the probability ratio of selecting that token:

$$r_m(\theta) = \frac{\text{LM}_\theta (z_m \mid z_{<m}, X)}{\text{LM}_{\theta_{\text{old}}} (z_m \mid z_{<m}, X)} = \frac{\alpha_{m,j}}{\alpha_{m,j}^{\text{old}}}. \quad (5)$$

**Inference.** After GRPO training, we apply the multi-token sampling procedure at each of the first $m - 1$ steps to form the continuous token via the average of $K$ sampled embeddings.

## 5.2 Dirichlet Sampling

In this section, we present another method for generating continuous tokens at each step by interpreting the model's output distribution $\alpha_t \in \Delta^{v-1}$ as concentration parameters of a Dirichlet distribution. We introduce a scaling hyperparameter $\gamma > 0$ and define the Dirichlet distribution with parameters $\gamma \alpha_t = (\gamma \alpha_{t,1}, \ldots, \gamma \alpha_{t,v})$. Without this scaling, directly using $\alpha_t$ as parameters often causes training instability, particularly when many $\alpha_{t,i}$ values are small. We then sample a point $\hat{\alpha}_t \in \Delta^{v-1}$ from the resulting distribution $\text{Dir}(\gamma \alpha_t)$. After sampling, we form the continuous token by mapping $z_t = \boldsymbol{E}^\top \hat{\alpha}_t \in \mathbb{R}^d$, which becomes the query for the next step. We denote the Dirichlet densities

| $K$ | Val. Acc. (%) | | Val. Entropy (SFT → SFT+GRPO) | | | |
|---|---|---|---|---|---|---|
|  | SFT | SFT+GRPO | token$_1$ | token$_2$ | token$_3$ | token$_4$ |
| 24 1 |  | 49.01 | 0.32 → 0.03 | 0.59 → 0.07 | 0.55 → 0.16 | 0.48 → 0.17 |
| 24 3 | 39.76 | 52.60 | 0.37 → 0.06 | 0.75 → 0.21 | 0.80 → 0.33 | 0.53 → 0.15 |
| 24 6 |  | 49.69 | 0.45 → 0.12 | 0.77 → 0.34 | 0.84 → 0.66 | 0.51 → 0.22 |
| 32 1 |  | 51.61 | 0.36 → 0.01 | 0.63 → 0.04 | 0.35 → 0.26 | 0.20 → 0.12 |
| 32 3 | 43.50 | 55.66 | 0.39 → 0.04 | 0.71 → 0.09 | 0.54 → 0.58 | 0.29 → 0.17 |
| 32 6 |  | 50.38 | 0.42 → 0.06 | 0.79 → 0.22 | 0.61 → 0.85 | 0.28 → 0.15 |

Table 2: Validation accuracy and token-level entropy of CoT2–MTS GRPO on a discrete CoT model across rollout sizes $K$ for MNNS. We use 4 input digits in 1–9; 1-layer, 1-head GPT2 with embedding 24 and 32; SFT accuracies are 39.76% and 43.50%, respectively.

induced by current and old policies as $f_\theta(\hat{\alpha}_t; \gamma \alpha_t)$ and $f_{\theta_{\text{old}}}(\hat{\alpha}_t; \gamma \alpha_t^{\text{old}})$, respectively. Accordingly, we define the policy ratio at a continuous step $t < m$ as:

$$r_t(\theta) = \frac{\text{LM}_\theta(z_t \mid z_{<t}, X)}{\text{LM}_{\theta_{\text{old}}}(z_t \mid z_{<t}, X)} = \frac{f_\theta(\hat{\alpha}_t; \gamma \alpha_t)}{f_{\theta_{\text{old}}}(\hat{\alpha}_t; \gamma \alpha_t^{\text{old}})},$$

The above definition parallels the probability ratio for discrete actions, but replaces the categorical pmf with a continuous Dirichlet pdf. At the final step $t = m$, we sample a discrete token $z_m \in \{e_1, \ldots, e_v\}$ from $\alpha_m$, and use the standard policy ratio in (5). At inference, we follow the aforementioned autoregressive procedure by forming $z_t = \boldsymbol{E}^\top \alpha_t$.

### 5.3 RESULTS AND DISCUSSION OF POLICY OPTIMIZATION FOR CoT2

**MNNS evaluation:** Table 2 demonstrates that, for each $K \in \{1, 3, 6\}$, CoT2-MTS significantly improves validation accuracy relative to the discrete SFT baseline (39.76%), with moderate $K$ yielding the best performance. We also observe that smaller $K$-values correspond to larger reductions in token-level entropies, suggesting that the model becomes more confident in each intermediate step by learning to commit to fewer tokens. This suggests a curriculum on $K$—starting small and gradually increasing—could potentially further improve the training on the MNNS task. Interestingly, the third token's entropy remains relatively high, which might indicate that the model hedges among partial expansions at this step that preserves useful diversity. Therefore, CoT2-MTS enables a discrete CoT model to produce continuous outputs and improves its final performance.

**ProsQA and ProntoQA evaluation:** Table 1 shows the benefits of GRPO with CoT2-MTS on models trained with discrete or continuous SFT. **Remarkably, both CoT2 and discrete CoT models consistently improve across all rollout sizes** ($K = 6, 8, 12$), with larger $K$ values yielding better final accuracies by promoting more exploration. Notably, the discrete CoT benefits more from RL training compared to CoT2, likely because the CoT2 already internalizes exploration through CSFT training. Aligning with this, for ProntoQA task, the final performance of discrete CoT approaches that of the CoT2 model. Finally, gains on MNNS are smaller than on ProsQA/ProntoQA, likely because MNNS is highly structured and closely aligned with the CSFT targets, leaving limited headroom for RL.

In Appendices C.4 and D.2, we provide additional results showing that GRPO with Dirichlet sampling further improves CSFT-trained CoT2. We also report preliminary GRPO results on GSM8K with Qwen3-0.6B, where CoT2-MTS consistently improves over discrete CoT while reducing response length, suggesting that exploration in the continuous action space via MTS is a promising direction.

### 5.4 RELATED WORK

The efficacy of eliciting reasoning in LLMs through chain-of-thought (CoT) prompting has been well-established (Nye et al., 2021; Wei et al., 2022; Kojima et al., 2022; Suzgun et al., 2023; Guo et al., 2025). CoT prompting conveniently increases inference-time compute and computational depth, both of are independently useful (Pfau et al., 2024; Goyal et al., 2024; Feng et al., 2023; Merrill & Sabharwal, 2024). However, the discrete nature of CoT tokens forces sequential exploration of reasoning paths, resulting in longer reasoning paths and consequently increased inference-time compute. Furthermore, restricting reasoning to natural language can be inefficient, as groups of tokens can often be more effectively represented by a single continuous token. Thus, CoT2 offers an alternative strategy for compute-efficient reasoning and complements methods that aim to shorten/control the trace length of CoT (Aggarwal & Welleck, 2025; Zhang et al., 2025a; Sui et al., 2025).

Our work is most related to a recent body of work introducing LLMs capable of reasoning with explicit continuous tokens decoded autoregressively. In particular, recently proposed COCONUT (Hao et al., 2024) autoregressively feeds the last token's final-layer representation as input to the next step. Given labeled CoT data, COCONUT is trained to progressively replace discrete tokens with continuous tokens (from left to right). While COCONUT and our CoT2 aim to reason in continuous space, we propose distinct algorithmic approaches that also address the exploration challenge. Key differences include: (1) Our continuous tokens are simplex-weighted compositions of vocabulary tokens. (2) Our supervision method is novel and explicitly targets implicit parallelism. (3) CoT2 does not initialize from, nor attempt to mimic, discrete CoT. (4) By introducing sampling strategies and associated GRPO variations, we realize the "Supervised Training → Reinforcement Learning" paradigm in the context of CoT2. A concurrent work Zhu et al. (2025) provides a theoretical construction for a continuous chain of thought on graph reachability problems, which is at a high level similar to our supervision CSFT. We provide further discussion of the literature on multi-token prediction and reinforcement learning in Appendix B.

## 6 CONCLUSION

In this paper, we provided a thorough theoretical characterization of CoT2 spanning the achievable level of parallelism and dimension-budget tradeoffs, CoT2-specific capabilities of transformer, and the statistical benefits of CoT2. Our theory is developed in tandem with novel continuous supervision (CSFT) and reinforcement learning strategies. As future work, applying CSFT selectively on segments of the LLM trajectories represents a promising direction for equipping LLMs with CoT2 capabilities.

ACKNOWLEDGEMENTS

This work is supported by the National Science Foundation grants CCF-2046816, CCF-2403075, CCF-2212426, the Office of Naval Research grant N000142412289, a gift from Open Philanthropy, and an Adobe Data Science Research Award. The computational aspects of the research are generously supported by computational resources provided by the Amazon Research Award on Foundation Model Development and by the Google Cloud Research Credits program with the award GCP440569534 on Continuous Chain-of-Thought Models.

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

# APPENDIX

We discuss additional related work in Appendix B. We provide further implementation details in Appendix C, including those for the MNNS task (Appendix C.1), the ProntoQA/ProsQA datasets (Appendix C.2), and GRPO training (Appendix C.4). We present additional experimental results in Appendix D, and we offer details on continuous supervised training and GRPO in Appendix D.1 and Appendix D.2, respectively. Finally, we include the proofs of Propositions 1, 2, and 3 in Appendix E.

## A    LLM USAGE

We used large language models solely as assistive tools. They provided support for text clarity/grammar, coding utilities, LaTeX edits, and literature search. The LLM did not design algorithms, implement core methods, choose hyperparameters, or run experiments. All scientific claims, figures, math, and code were authored and verified by the authors.

## B    FURTHER RELATED WORK

One way to address the limitation of discrete tokens is by leveraging the implicit reasoning capabilities of transformers (Yang et al., 2024; Shalev et al., 2024). Works such as (Deng et al., 2023; 2024; Yu et al., 2024) use various techniques to obtain models that can perform reasoning internally without emitting CoT tokens. Another line of work has found looped transformers to be effective on reasoning problems (Giannou et al., 2023; Geiping et al., 2025), notably being able to mimic CoT (Saunshi et al., 2025) with a sufficient number of iterations. Another way to address these challenges is to incorporate explicit continuous tokens similar to COCONUT (Hao et al., 2024). Specifically, Shen et al. (2025) proposes CODI, where an LLM with continuous CoT is supervised to produce the correct answer, while also aligning its hidden representation on the last reasoning token to that of a discrete CoT model that shares the same backbone. Cheng & Van Durme (2024) propose CCOT, where an auxiliary module is first trained to decode autoregressively a compressed representation of a discrete CoT trace, and later the main LLM is fine-tuned to produce correct answers by additionally conditioning on the generated continuous tokens. Our work is similar to this line of work in that continuous representations are used to perform reasoning. In parallel with our work, Zhang et al. (2025b) propose Soft Thinking which is a training-free inference scheme that replaces intermediate discrete tokens with softmax-weighted concept tokens and uses an entropy-based early stop; unlike CoT2, we introduce explicit continuous supervision (CSFT) and RL with multi-token sampling (MTS/GRPO) to track multiple trajectories with controllable parallelism.

The proposed CoT2 approach simultaneously tracks all possible trajectories and superposes them within continuous tokens. This approach is similar to that of Xiong et al. (2025), who superpose multiple candidate outputs into a single final token. Our approach also shares similarities with decoding algorithms like self-consistency (Wang et al., 2022) and Best-of-N-Sampling (Stiennon et al., 2022), which generate multiple trajectories by running inference multiple times and then select a final answer based on the aggregate statistics. In contrast, our algorithm performs a single inference, superposing different trajectories all at once and determining the final answer in one pass. A concurrent work by Yue et al. (2025) explores hybrid latent RL using a learnable gating mechanism, where continuous tokens are directly linked to model parameters and thus each trajectory can only be used for a single gradient update due to on-policy constraints. Whereas, our MTS strategy integrates stochasticity within continuous tokens and makes exploration possible through continuous tokens. Furthermore, our Dirichlet sampling approach for generating multiple rollouts in GRPO training draws connections to previous works such as Latent Dirichlet Allocation (LDA) (Blei et al., 2003), which introduces Dirichlet priors within a hierarchical Bayesian framework, and AlphaGo (Silver et al., 2017), which injects Dirichlet noise to encourage exploration.

Our work also tangentially relates to research on multi-token prediction (Bachmann & Nagarajan, 2024; Liu et al., 2024; Gloeckle et al., 2024), which aims to improve the efficiency and quality of generation by predicting multiple tokens at once. It is hypothesized that effective future prediction necessitates the exploration of many possible continuations, which is similar to our CoT2 approach.

# C  IMPLEMENTATION DETAILS

**Computational Resources:** All experiments were run on a Slurm-managed cluster using L40S GPUs with 48GB of memory. Each experiment fits on a single GPU. In the case of 4 input digits, the SFT or CSFT training takes approximately 3 hours on a single GPU. For 5-digit inputs, the dataset size increases by roughly a factor of 10, and the training time increases proportionally. The entire codebase was implemented in PyTorch.

**Remark 1** (Scalability of base CoT2 decoding.). *Computing the continuous token $z_t = E^\top \alpha_t$ adds a single additional $O(vd)$ matrix-vector multiply per step. This matches the $O(vd)$ cost already incurred by the standard output projection that produces the logits $\ell_t = W_{out} h_t$ and distribution $\alpha_t = \text{softmax}(\ell_t)$. Moreover, this additional cost is independent of the context length $N$ and is dominated by the transformer stack's per-token compute, so in practice the overhead is minor even for larger language models.*

## C.1  IMPLEMENTATION DETAILS OF EXPERIMENTS ON MNNS TASK

**Dataset Details:** For the MNNS task, the vocabulary consists of a range of numbers from $[-S, S]$ for some positive integer $S$, together with *<BOS>*, *<EOS>*, and $\rightarrow$ special tokens. The integer $S$ is chosen so that all possible partial sums of the selected input digits lie within $[-S, S]$. For example, when the input digits lie in the range 1–10, we set $S = 36$, whereas for digits in 5–14, we set $S = 40$. We performed our experiments on the 4 and 5 input digit scenarios. A sample input line with $m$ numbers is:

$$\text{<BOS>}\, D_1\, D_2\, \ldots\, D_m \rightarrow$$

Accordingly, the output will be $m$ sum tokens, where the final token corresponds to the answer, followed by *<EOS>* token:

$$S_1\, S_2\, \ldots\, S_m\, \text{<EOS>}$$

As a concrete example, consider the input $2, 1, 4$ ($m = 3$), following Figure 1. In this case, the solution for the MNNS task is $-2 - 1 + 4 = 1$. Therefore, for the discrete model, the input is *<BOS>* $D_2\, D_1\, D_4 \rightarrow$ and we supervise it along the trajectory of correct output tokens $S_{-2}\, S_{-3}, S_1 \text{<EOS>}$, as illustrated in Figure 1. On the other hand, the continuous supervision at the first step holds $S_2$ and $S_{-2}$ as possibilities. Then, for the next step, we add 1 or -1 to these numbers, and the resulting possibilities are $S_3, S_1, S_{-1}, S_{-3}$. Finally, at the last step, the model is supervised to pick the correct answer $S_1$ as the token.

We split the datasets by ensuring that each permutation of a set of numbers is exactly in one of the train and validation datasets, as the answer to the question is permutation-invariant. This way, we prevent the models from memorizing the answer and make a fair comparison. We also use 0.8-0.2 split for train-val datasets.

**Model and Hyperparameters:** We use the GPT2 model, with 1 layer 1 head, 2 layer 2 head, and 4 layer 4 head as the configurations. For each configuration, we experiment with embedding dimensions of 16, 24, or 32. We train with a learning rate of $\texttt{lr} = 10^{-4}$ and use AdamW (no weight decay). The batch size is 16 for 4-digit inputs and 64 for 5-digit inputs.

**Evaluation of the models:** To make a proper comparison, we only check the final answer of the models, as checking the correctness of the full path of the discrete model would be unfair.

**Pass@k Experiment:** We perform our experiments for temperatures 0, 0.4, 0.8, and 1 by repeating the evaluation 10 times for each $k$ value where k changes from 1 to 14.

## C.2  IMPLEMENTATION DETAILS OF EXPERIMENTS ON PRONTOQA/PROSQA DATASETS

**Dataset Details:** Different from the original ProntoQA/ProsQA datasets which described the structured ontology in natural language as a set of known conditions, we use a more structured format through a token-level representation. An example prompt is shown below.

**Description of the structured ontology:** Each component of the ontology and associated questions is represented through discrete tokens with their own learned embeddings, rather than as raw textual input. Specifically, we use the GPT2 architecture and encode the ontology's structural components.

Below are two examples demonstrating how natural-language assertions are mapped to our tokenized format:

`Brimpuses are not luminous → 'A' 'not in' 'B' '.'.`

`Shumpuses are amenable; Each yumpus is a lorpu; Every lorpus is floral →` `'C' 'in' 'D' '.'.`

Below, we have the ProntoQA and ProsQA datasets' input-output format.

**The structure of ProntoQA:**

**Input:**`'Description' '{' 'A' 'not in' 'B' '.' ... 'C' 'in' 'D' '.' '}'` `'Question' '{' 'C' 'not in' 'F' '.' '}'`

**Output:**`'Steps' { 'C' 'in' 'D' '.' ... 'D' 'in' 'E' '.' '}' 'Answer' '{'` `'False' '}'`

**The structure of ProsQA:**

**Input:**`'Description' '{' 'A' 'in' 'B' '.' ... 'C' 'in' 'D' '.' '}'` `'Question' '{' 'C' 'in' 'F' 'or' 'E' '}'`

**Output:**`'Steps' { 'C' 'in' 'D' '.' ... 'D' in 'E' '.' '}' 'Answer' '{' 'F'` `'}'`

Each distinct component or relation (e.g., `'A'`, `'in'`, `'not in'`) is treated as a unique token, and singular/plural variants (such as `'lempus'` and `'lempuses'`) are collapsed into a single token to simplify the vocabulary. Alongside these concept tokens, special structural tokens (`'Description'`, `''`, `''`, `'.'`, `'or'`, etc.) are also included, which results in a vocabulary size of 31 tokens. To avoid biases, we balance the dataset. In ProntoQA, "yes" and "no" each appear with 50% probability, and in ProsQA, the correct answer is randomly permuted at the first or second position. For all the other experimental and training settings, we follow Hao et al. (2024).

**Model and Hyperparameters:** We use the GPT2 model, with 2 layer 2 head, and 4 layer 4 head as the configurations. We tested embedding dimensions $24, 32, 40$ with these configurations. We set batch size 64. We train with a learning rate of $10^{-4}$ and use AdamW (no weight decay).

**Maj@k Experiment:** We use majority voting for evaluation instead of Pass@k, because both ProntoQA and ProsQA are binary questions. We perform our experiments for temperatures 0, 0.4, 0.8, and 1 by repeating the evaluation 10 times for each $k$ value where k changes from 1 to 21. If two or more answers end up with the same top vote, we pick one randomly.

### C.3 Implementation Details of Baselines

In Figure 2, we evaluate the following baselines under matched total number of epochs:

**Discrete no-CoT.** The model is trained to predict the final answer directly, without intermediate thought tokens.

**Discrete CoT.** The model is supervised on the full, correct path during SFT training.

**COCONUT.** Using a left-to-right curriculum training, the thought token at step $t$ is replaced by the model's last hidden state sequentially. During inference, the model outputs the answer at the final step ($t = m$) after $m - 1$ continuous thought tokens. To keep the evaluation fair, we use the same total number of epochs across all baselines. For COCONUT's $m$ curriculum stages, we divide the total epochs evenly across curriculum stages; e.g., with 1000 epochs and $m = 5$ output tokens, we train 200 epochs for each stage.

### C.4 Implementation Details of GRPO Training

In Hao et al. (2024), the reference model is updated by $\text{LM}_{\theta_{\text{ref}}} \leftarrow \text{LM}_{\theta}$ in each iteration (epoch). This approach is reasonable for their setting with a large dataset and a small number of epochs over it. For our setting, however, we set the reference model to the initial model and never update it through

---

**Algorithm 1** Multi-Token Sampling GRPO for Continuous Token Generation

---

**Input:** Initial policy $\text{LM}_{\theta_{\text{init}}}$; hyperparameters $K, G, m, \epsilon, \beta$.

1: $\text{LM}_\theta, \text{LM}_{\theta_{\text{ref}}} \leftarrow \text{LM}_{\theta_{\text{init}}}$
2: **for** iteration $= 1, 2, \ldots, I$ and **for** step $= 1, 2, \ldots, S$ **do**
3:     Sample a batch of inputs $\{X^{(b)}\}_{b=1}^{B}$
4:     Update $\text{LM}_{\theta_{\text{old}}} \leftarrow \text{LM}_\theta$
5:     **for** each input $X$ in the batch and **for** each trajectory $i = 1, \ldots, G$ from that $X$ **do**
6:         **for** each token step $t = 1, \ldots, m$ **do**
7:             **if** $t < m$ **then**                                                 ▷ Continuous token
8:                 Sample $K$ tokens $\{e_{i_1}, \ldots, e_{i_K}\}$ from $\alpha_t^{(i),\text{old}}$ to create continuous token $z_t \leftarrow \frac{1}{K} \sum_{r=1}^{K} e_{i_r}$.
9:                 Policy ratio for continuous token $r_t(\theta) \leftarrow \left( \prod_{r=1}^{K} \alpha_{t,i_r}^{(i)} / \prod_{r=1}^{K} \alpha_{t,i_r}^{(i),\text{old}} \right)^{\frac{1}{K}}$.
10:             **else**                                                       ▷ Final discrete token
11:                 Sample $z_m = e_j$ from $\alpha_m^{(i),\text{old}}$.
12:                 Policy ratio for discrete token $r_m(\theta) \leftarrow \alpha_{m,j}^{(i)} / \alpha_{m,j}^{(i),\text{old}}$.
13:     Obtain advantage estimates $\hat{A}_{i,t}$ for each token $t$ in each trajectory $Z^{(i)}$ and calculate objective.
14:     Update $\theta$ to minimize $\mathcal{L}_{\text{GRPO}}(\theta)$.

**Output** $\text{LM}_\theta$

---

iterations as we have a smaller dataset. Meanwhile, we update the old model before every batch $\text{LM}_{\theta_{\text{old}}} \leftarrow \text{LM}_\theta$.

In our experiments, we use $G = 8$ trajectories per input data point, use clipping parameter $\epsilon = 0.1$, and set the KL-divergence coefficient $\beta = 0$ in most cases (with $\beta = 0.1$ in a few). For the CoT2 model with MTS sampling, we change the number of tokens to sample $K$ from 1 to 12. In the MNNS task, the 5-digit case has about ten times more data than the 4-digit case, so we typically focus on 4-digit MNNS because of computational considerations and use a batch size of 16 in those experiments.

Learning rates differ by model and setting. We use $\texttt{lr} = 5 \times 10^{-5}$ for CoT2-MTS sampling (figures in the main text), $\texttt{lr} = 1 \times 10^{-5}$ for discrete CoT with Dirichlet sampling, and $\texttt{lr} = 1 \times 10^{-6}$ for CoT2 with Dirichlet sampling. For ProntoQA and ProsQA experiments, we perform a grid search over learning rates ranging from $1 \times 10^{-4}$ to $1 \times 10^{-8}$ and select and report results using the best-performing configuration. For most settings, we find $\texttt{lr} = 1 \times 10^{-5}$ optimal; however, for CoT2 and discrete CoT models with $K = 6$, we set $\texttt{lr} = 1 \times 10^{-6}$. We also use AdamW with a weight-decay of 0.01. For Dirichlet experiments on the MNNS task, we try various scale parameters $\gamma$, but we find $\gamma = 20$ to work best in most settings. Unless stated otherwise, we report the best validation accuracy found during training for each setting.

### C.4.1 Further Details on Multi-Token Sampling

The full algorithm describing MTS sampling is provided in Algorithm 1.

**Remark.** An alternative to the normalization in (4) is scaling the logits by $1/K$ before applying softmax. However, this leads to a distribution shift from the SFT-trained model at inference, and ultimately degrades performance.

## D  Experimental Results

### D.1 Continuous Supervised Training Results

**Teacher Forcing and Self-feeding Comparison:** As described in CSFT section, we tested two approaches of providing prefixes during training the CoT2 model with CSFT. Although the model autoregressively generates at inference time, teacher forcing yields better performance than self-feeding during CSFT training. We also tested curriculum settings, where we switch to self-feeding after a pre-determined number of epochs in the training. Still, the accuracies didn't improve beyond pure teacher-forcing training. The results are illustrated in Figure 6, where we refer to teacher-forcing as "hard-teacher" and refer to self-feeding as "soft-teacher".

**Sparse Supervision for Discrete Baseline:** We also tested providing a subset of the correct path to the discrete model. We observed that a sparsely supervised discrete model can achieve better

| Method | MNNS | ProsQA | ProntoQA |
|---|---|---|---|
| CoT2 | 98.94 | 93.37 | 98.01 |
| COCONUT | 92.58 | 90.03 | 96.94 |
| Discrete CoT | 84.92 | 68.50 | 82.47 |
| Discrete no-CoT | 68.35 | 54.91 | 73.65 |

Table 3: Validation accuracies on MNNS, ProsQA, and ProntoQA for CoT2, COCONUT, Discrete CoT, and Discrete no-CoT. **Setting:** MNNS with 4 input digits in 1–9; CoT2 uses full budget $B$ and Discrete CoT uses $B=1$. MNNS: 2-layer, 2-head GPT2 ($d=32$); ProsQA & ProntoQA: 4-layer, 4-head GPT2 ($d=32$).

performance than the fully supervised discrete model when the distribution is "easier" to handle by the model. As an example, we tested the case when we have 5 input digits from the range of 11 to 19. In this case, in nearly all of the cases, the answer to our MNNS game is (sum of minimum 3 numbers) - (sum of maximum 2 numbers) out of the 5 input numbers. In this case, when only 1 token from the correct path is provided to the discrete model, it's better than 3 and 5 token cases. However, when we change the distribution to a range of numbers from 5 to 13, which makes the question reasonably harder, the discrete model with 1 token supervision performs worse than the other two, and the discrete model with full supervision performs best. The results are demonstrated in Figure 4.

**Further Results on CoT2 vs Discrete CoT:** The results in Figure 7 also indicate that above an embedding dimension threshold, the CoT2 model has superior performance and trains significantly faster than the discrete CoT model. Moreover, combining the results of Figure 7 with Figure 3, we see that the CoT2 model with one layer and one head GPT2 model performs better than discrete CoT model with two layers and two heads at embeddings 24 and 32. While the continuous approach requires greater embedding capacity to support its distributional representations at each step, it can outperform the discrete model using fewer layers and attention heads. Supporting the findings in Figure 2, Figure 8 illustrates that on the ProntoQA task, CoT2 consistently outperforms the discrete CoT baseline when the embedding dimension is above a threshold. Likewise, as depicted in Figure 9 and Figure 10, the discrete CoT model requires multiple samplings (Maj@k) to match the single-shot performance of CoT2 on both ProntoQA and ProsQA, which indicates that CoT2 model is more sample-efficient.

**Empirical Evidence of CoT2 Alignment with Supervision During Inference:** We provide direct empirical evidence supporting that well-trained CoT2 models indeed follow the intended search patterns prescribed by continuous supervision (CSFT). Specifically, we analyze token-level entropy as a probe for model behavior during inference on the MNNS validation set. At each intermediate reasoning step $t$, CSFT ideally prescribes a uniform distribution over $2^t$ partial sums, yielding an entropy $H_t = t \ln 2$. On the MNNS validation set with four digits we measure token-level entropies obtained from inference with 4 input digits (1–9) using a 1-layer, 1-head GPT2 architecture at embedding dimension $d = 32$. The measured entropies at each intermediate step are:

$$H_1 = 0.6896, \ H_2 = 1.3682, \ H_3 = 1.9588, \ H_4 = 0.2461.$$

Observe that the token-level entropies closely match expected theoretical values: $H_1 = 0.6896 \approx \ln 2$, $H_2 = 1.3682 \approx \ln 4$, $H_3 = 1.9588 \approx \ln 8$, and a sharp drop at $H_4 = 0.2461$. The alignment for steps $t \leq 3$ indicates that the model maintains an approximately uniform superposition over roughly $2, 4, 8$ trajectories, rather than collapsing into a single path or averaging them. The steep entropy reduction at the final step ($t = 4$) clearly reflects the one-hot supervision on the final answer. This entropy pattern provides direct empirical evidence that CoT2 inference aligns precisely with the CSFT training objective: the model explores multiple reasoning branches in parallel until committing decisively to the correct outcome at the final step.

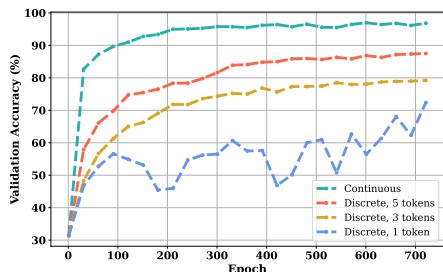

Figure 4: The figure illustrates that when the range of digits makes the question non-trivial on an MNNS task, the discrete CoT model trained with full token supervision outperforms sparse supervisions; in particular, single token supervision yields the worst performance. **Setting:** 5 input digits in $5 - 13$; 2-layer, 2-head GPT2 with $d = 32$.

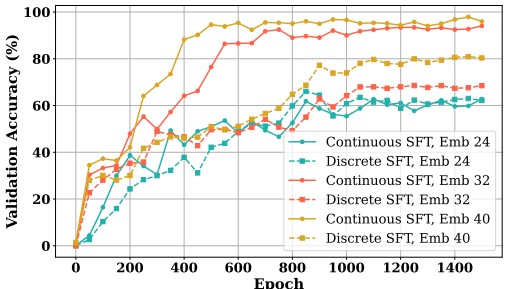

Figure 5: The figure reveals that CoT2 model is superior to discrete CoT in ProsQA, while also exhibiting faster convergence. **Setting:** 4-layer, 4-head GPT2 with $d \in \{24, 32, 40\}$.

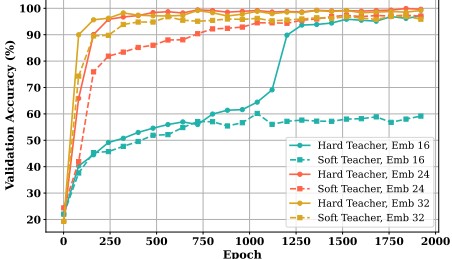

Figure 6: The comparison between the hard and soft teachers for different embedding dimensions on MNNS task. The figure illustrates that the hard teacher is superior to the soft teacher. **Setting:** 4 input digits in $1 - 9$; 4-layer, 4-head GPT2 with $d \in \{16, 24, 32\}$.

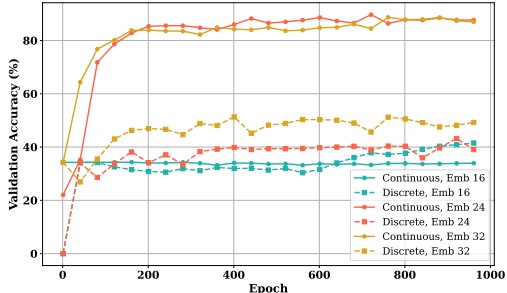

Figure 7: Comparison between CoT2 and discrete CoT model for different embedding dimensions. The figure demonstrates that above a certain embedding dimension threshold, the CoT2 model outperforms the discrete CoT model in the MNNS task. **Setting:** 4 input digits in $1 - 9$; 1-layer, 1-head GPT2 with $d \in \{16, 24, 32\}$.

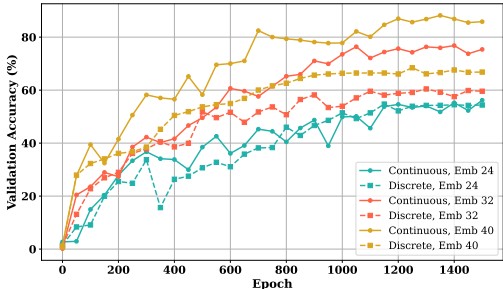

Figure 8: Comparison between CoT2 and discrete CoT model for different embedding dimensions in ProntoQA task. The figure shows that above an embedding dimension threshold, the CoT2 model outperforms the discrete CoT model. **Setting:** 4-layer, 4-head GPT2 with $d \in \{24, 32, 40\}$.

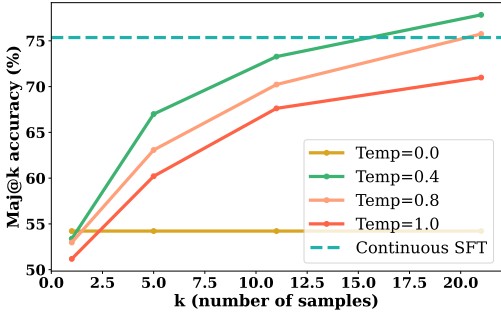

Figure 9: The figure illustrates that the discrete CoT model requires multiple samplings (Maj@k) to match the single-shot performance of the CoT2 model on ProntoQA (10-run average). **Setting:** 4-layer, 4-head GPT2 with $d = 32$.

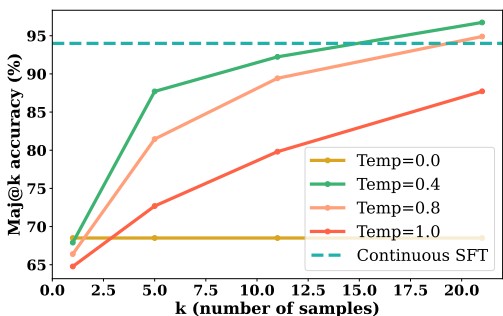

Figure 10: The figure illustrates that the discrete CoT model requires multiple samplings (Maj@k) to match the single-shot performance of the CoT2 model on ProsQA (10-run average). **Setting:** 4-layer, 4-head GPT2 with $d = 32$.

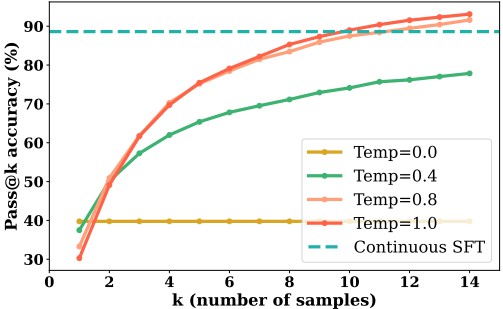

Figure 11: Performance of single-shot CoT2 vs. discrete CoT with Pass@$K$ across different temperatures on the MNNS task. **Setting:** 4 input digits in 1–9; 1-layer, 1-head GPT-2 with $d = 24$; CoT2 uses full trajectory budget $B=|\mathcal{T}|$ and discrete CoT uses $B=1$.

D.2   GRPO RESULTS

**Discussion on ProntoQA/ProsQA Datasets:** Table 1 illustrates that GRPO training using CoT2-MTS sampling consistently improves discrete CoT and CoT2 models over their initial SFT accuracy. Moreover, we observe that the improvement in the discrete CoT model is greater, which might indicate that the CoT2 model already gains an RL-like exploration mechanism through CSFT training. We observe that while increasing $K$ initially increases the accuracy by sampling more tokens at each step, beyond some $K$, the improvements diminish. This observation is consistent with Table 2, where we see that a moderate $K$ value offers the best final performance. One possible explanation is that while higher $K$ promotes better exploration, it also raises the chance of sampling unhelpful tokens that disrupt the averaged token representation. Indeed, for larger $K$, we observe that the RL objective saturates to near zero which suggests that most rollouts fail once the averaged token contains too many distracting tokens.

**Discussion on Dirichlet Sampling:** We also investigate the effects of Dirichlet sampling in GRPO training discrete CoT and CoT2 models. The results in Table 4 indicate that applying Dirichlet sampling ($\gamma = 20$) in GRPO training of discrete CoT model consistently improves over the initial SFT training accuracies. Similar to the CoT2 + MTS sampling results in Table 2, we observe that the entropy at the third token remains relatively high, which suggests a beneficial diversity in model's predictions for that token. Moreover, the Table 5 demonstrates that Dirichlet sampling also improves the CoT2 model's SFT accuracy, even though it has a high initial SFT accuracy. As illustrated in Table 5, we find there is an optimal value for the scale parameter $\gamma$, since larger $\gamma$ typically yields more uniform sampling distributions, whereas smaller $\gamma$ concentrates the distribution more sharply. Thus, adjusting $\gamma$ provides a balance between exploration and stability in GRPO training.

| Layers | Heads | Val. Accuracy (%) | | Val. Entropy (SFT → SFT+GRPO) | | | |
|---|---|---|---|---|---|---|---|
| | | SFT | SFT+GRPO | $\text{token}_1$ | $\text{token}_2$ | $\text{token}_3$ | $\text{token}_4$ |
| 1 | 1 | 39.76 | 46.25 | $0.4851 \to 0.1701$ | $0.5165 \to 0.6380$ | $0.3243 \to 0.6590$ | $0.1597 \to 0.4878$ |
| 2 | 2 | 70.26 | 75.84 | $0.4851 \to 0.4027$ | $0.5165 \to 0.4413$ | $0.3243 \to 0.2907$ | $0.1597 \to 0.1386$ |

Table 4: Discrete CoT models trained with GRPO after SFT using Dirichlet sampling ($\gamma = 20$) and a learning rate of $1 \times 10^{-5}$. We show validation accuracy (%) and token-level entropy (SFT → SFT+GRPO) for each (Layers, Heads) setting, with an embedding dimension of 24 for GPT2 model.

| Dirichlet Scale ($\gamma$) | SFT Val. Acc (%) | SFT + GRPO Val. Acc (%) |
|---|---|---|
| 10 | 87.84 | 89.76 |
| 20 | 87.84 | 90.75 |
| 40 | 87.84 | 90.37 |

Table 5: Validation accuracies GRPO training with CoT2 models using different Dirichlet sampling scales ($\gamma$) with learning rate of $1 \times 10^{-6}$. We show the baseline SFT accuracy (87.84%) and final performance after GRPO.

**CoT2-MTS GRPO experiments on GSM8K.** According to raised points to evaluate our CoT2-MTS method on reasoning benchmarks beyond logical multi-hop reasoning, we ran Qwen3-0.6B on GSM8K using our CoT2-MTS continuous-token sampling procedure. We implemented the continuous rollout/training code from scratch, and to keep generation and training computationally feasible, we conducted our experiments on modest model sizes and decoding budgets. For CoT2-MTS, we used a curriculum learning approach where continuous tokens are introduced from 50% of training, ramped to full by 80%, and used thereafter. The table below reports single-run results for different maximum response lengths (160, 128, 96, 80 tokens). We use 16 continuous tokens for the 96/80 settings and 32 continuous tokens for the 128/160 settings; for example, in the 160-token configuration the effective budget grows from 128 discrete tokens at the start of training to 128 discrete + 32 continuous tokens (i.e., 128→160) as the curriculum completes.

For these experiments we adopt a sparse-reward setting by using only the final answer in the 0/1 reward, and train with GRPO for 2000 steps using learning rate $3 \times 10^{-6}$, batch size 6, and gradient accumulation of 4. The GRPO group size is set to $G = 4$, the clipping parameter to $\epsilon = 0.1$, the KL coefficient to 0, and the number of sampled tokens in MTS to $K = 4$.

| Max Resp. Len | Initial Acc (%) | Disc. CoT Acc (%) | CoT2-MTS Acc (%) | Avg. Len (Disc CoT) | Avg. Len (CoT2-MTS) |
|---|---|---|---|---|---|
| 160 | 34.6 | 37.0 | **41.3** | **97.3** | 114.8 |
| 128 | 27.6 | 34.3 | **36.2** | 104.8 | **96.6** |
| 96 | 17.2 | **27.2** | 27.1 | 84.0 | **83.7** |
| 80 | 9.5 | 21.5 | **27.0** | 73.9 | **67.0** |

Table 6: Validation accuracies and response lengths for GRPO training with CoT2-MTS ($K = 4$) on GSM8K using Qwen3-0.6B under different maximum response lengths.

Across GSM8K with Qwen3-0.6B, CoT2-MTS improves over discrete CoT in three out of four maximum response-length settings and essentially matches it in the remaining one, while often shortening the average response length. Overall, these preliminary results indicate that performing exploration in the continuous action space via CoT2-MTS is a promising direction.

## E  THEORETICAL DETAILS

This section of the appendix gathers theoretical results that support our continuous chain-of-thought framework. Appendix E.1 presents a simple packing-style argument that makes the tradeoff between the trajectory budget (i.e., the number of superposed states) and the embedding dimension needed for robust representation explicit. Appendix E.2 discusses when Assumption 1 is a reasonable modeling assumption for structured reasoning tasks. In Appendix E.3, we provide proofs for our consistency and sample-complexity comparisons between CoT2-MTS and discrete CoT. Finally, Appendix E.4 gives an explicit transformer construction that solves the MNNS task.

### E.1  DISCUSSION ON BUDGET-EMBEDDING DIMENSION TRADEOFF

Without loss of generality, assume that the token embeddings lie inside unit-circle $\|e_i\|_2 \leq 1$. Let $S \subseteq [v]$ be a set of superposed states at a step with size $|S| = B$. By triangle-inequality, we know that the convex superposition $z_S = \sum_{i \in S} \alpha_i e_i$ with $\alpha_i \geq 0$ and $\sum_{i \in S} \alpha_i = 1$ satisfies $\|z_S\|_2 \leq 1$.

Suppose there exists a downstream module to recover the superposed states $S$ robustly from the single $d$-dimensional vector $z_S$. Formally, assume a fixed margin $\delta > 0$ such that even if $z_S$ is perturbed by any $\eta$ with $\|\eta\|_2 \leq \delta$, the decoding does not change: there is a decoder $\text{Dec} : \mathbb{R}^d \to \{S \subseteq [v] : |S| = B\}$ with $\text{Dec}(z_S + \eta) = S$ for all $S$ and all such $\eta$.

For each $z_S$, consider the closed ball $B(z_S, \delta) = \{x : \|x - z_S\| \leq \delta\}$. The robustness assumption trivially implies these balls are pairwise disjoint; moreover, by the triangle inequality, for any $x \in B(z_S, \delta)$,

$$\|x\| \leq \|x - z_S\| + \|z_S\| \leq \delta + 1,$$

so, each $\delta$-ball sits inside a larger $1 + \delta$ ball $B(z_S, \delta) \subseteq B(0, 1 + \delta)$. Therefore, the union of all disjoint $\delta$-balls lies inside $B(0, 1 + \delta)$:

$$\bigcup_{\substack{S \subseteq [v] \\ |S| = B}} B(z_S, \delta) \subseteq B(0, 1 + \delta).$$

The number of different sets of superposed states of size $B$ is $\binom{v}{B}$. The packing constraint says the union of the $\binom{v}{B}$ disjoint $\delta$-balls must fit inside $B(0, 1 + \delta)$. In $\mathbb{R}^d$, the volume of a radius-$r$ Euclidean ball is $c_d r^d$ where $c_d = \frac{\pi^{d/2}}{\Gamma(1+d/2)}$. Comparing volumes gives

$$\binom{v}{B} \leq \frac{\text{vol}(B(0, 1 + \delta))}{\text{vol}(B(0, \delta))} = \left(\frac{1+\delta}{\delta}\right)^d.$$

In the regime $v \gg B$, taking logs and using the Stirling approximation for $B!$ gives the inequality $\log\binom{v}{B} \geq B\log(v/B)$. Using this yields:

$$B\log\frac{v}{B} \leq d\log\left(\frac{1+\delta}{\delta}\right) \leq d\log\left(\frac{2}{\delta}\right)$$

for $\delta \leq 1$. Equivalently, $d \geq \frac{B\log(v/B)}{\log(2/\delta)} = \Omega(B\log(v/B))$ when $\delta$ is a fixed constant.

### E.2 Justification for Assumption 1

Assumption 1 holds for tasks where the next-token distribution depends solely on the current token and the input tokens rather than the full history of output tokens. This is satisfied by many reasoning tasks, where the aim is to keep track of an intermediate state (e.g., the current sum) and update this state based only on the current state and the input, independently of the earlier trajectory.

For example, in the MNNS task, the model generates a token representing the current partial sum at each step. To compute the distribution over the next possible sums, the model adds or subtracts the selected number from the input context $X$ to the current sum, without needing to remember the sequence of previous sums explicitly. Thus, the next-state distribution at each step is only determined by the current state and it naturally satisfies the Assumption 1.

### E.3 Proofs for Consistency and Sample Complexity Results for CoT2-MTS vs. Discrete CoT

**Proposition 2** (Consistency of CoT and CoT2 inference). *Under Assumption 1 and given $X$, the output of base CoT2 is $z_m = \sum_{j=1}^{v} \alpha_{m,j} e_j$ where $\alpha_m = \alpha_1 \prod_{t=1}^{m-1} M_t$. Discrete CoT and CoT2-MTS have the same output once we take the expectation over their stochastic sampling.*

*Proof.* Let $\hat{\alpha}_t^{(\text{disc})}, \hat{\alpha}_t^{(\text{MTS})}$ denote the empirical output token distributions at step $t$ under one trajectory obtained by the discrete CoT, and CoT2 with MTS models, respectively. We define $\alpha_t^{(\text{disc})} = \mathbb{E}\left[\hat{\alpha}_t^{(\text{disc})}\right]$ and $\alpha_t^{(\text{MTS})} = \mathbb{E}\left[\hat{\alpha}_t^{(\text{MTS})}\right]$ to be corresponding expected distributions. The discrete CoT model at each step picks exactly 1 token from $\alpha_t$. On the other hand, CoT2-MTS samples $K$ i.i.d. tokens at every step independently according to their probabilities from $\hat{\alpha}_t$. We denote them $i_1, \ldots, i_K$, and average their embeddings to produce a single query.

We will use induction in our argument. For the base case, all models start with the same initial distribution, so we trivially have $\alpha_1^{(\text{disc})} = \alpha_1^{(\text{MTS})} = \alpha_1^{(\text{CoT2})}$. For the inductive step, assume that we have $\alpha_{t-1}^{(\text{disc})} = \alpha_{t-1}^{(\text{MTS})} = \alpha_{t-1}^{(\text{CoT2})}$. We will show that $\alpha_t^{(\text{disc})} = \alpha_t^{(\text{MTS})} = \alpha_t^{(\text{CoT2})}$. On the other, for the discrete CoT model, the model samples one token $e_{i_{t+1}}$ from the row of $M_t$ for a token $i_{t+1}$. Therefore, we need to condition on the token at step $t$. We have:

$$\mathbb{E}\left[\hat{\alpha}_{t+1}^{(\text{disc})}\right] = \sum_{j=1}^{v} \mathbb{P}(z_t = e_j) \mathbb{E}\left[\hat{\alpha}_{t+1}^{(\text{disc})} \mid z_t = e_j\right] \tag{6}$$

$$= \sum_{j=1}^{v} \mathbb{P}(z_t = e_j) \text{LM}_\theta^{(t)}(\cdot \mid e_j, X)$$

$$\sum_{j=1}^{v} \alpha_{t,j}^{(\text{disc})} \text{LM}_\theta^{(t)}(\cdot \mid e_j, X)$$

$$\overset{(a)}{=} \sum_{j=1}^{v} \alpha_{t,j}^{(\text{CoT2})} \text{LM}_\theta^{(t)}(\cdot \mid e_j, X)$$

$$\overset{(b)}{=} \alpha_{t+1}^{(\text{CoT2})} \tag{7}$$

where (a) follows from the induction argument and (b) follows from Assumption 1. Therefore, we obtain $\alpha_{t+1}^{(\text{disc})} = \mathbb{E}\left[\hat{\alpha}_{t+1}^{(\text{disc})}\right] = \alpha_{t+1}^{(\text{CoT2})}$. For the CoT2-MTS model, the argument will be similar. Using the decoupling of trajectories by Assumption 1, the next distribution is:

$$\text{LM}_\theta^{(t)}\left(\cdot \mid \frac{1}{K} \sum_{r=1}^{K} e_{i_r}, X\right) = \frac{1}{K} \sum_{r=1}^{K} \text{LM}_\theta^{(t)}(\cdot \mid e_{i_r}, X).$$

Therefore, we write:

$$
\begin{aligned}
&\mathbb{E}\left[\hat{\boldsymbol{\alpha}}_{t+1}^{(\mathrm{MTS})}\right] \\
&= \sum_{(i_1,\ldots,i_K)\in[v]^K} \mathbb{P}(\boldsymbol{e}_{i_1},\ldots,\boldsymbol{e}_{i_K})\,\mathbb{E}\left[\hat{\boldsymbol{\alpha}}_{t+1}^{(\mathrm{MTS})}\mid \boldsymbol{e}_{i_1},\ldots,\boldsymbol{e}_{i_K}\right] \\
&= \sum_{(i_1,\ldots,i_K)\in[v]^K} \mathbb{P}(\boldsymbol{e}_{i_1},\ldots,\boldsymbol{e}_{i_K})\,\mathrm{LM}_\theta^{(t)}\!\left(\cdot\mid \frac{1}{K}\sum_{r=1}^K \boldsymbol{e}_{i_r},\boldsymbol{X}\right) \\
&= \sum_{(i_1,\ldots,i_K)\in[v]^K} \mathbb{P}(\boldsymbol{e}_{i_1},\ldots,\boldsymbol{e}_{i_K})\frac{1}{K}\sum_{r=1}^K \mathrm{LM}_\theta^{(t)}\left(\cdot\mid \boldsymbol{e}_{i_r},\boldsymbol{X}\right) \\
&= \sum_{(i_1,\ldots,i_K)\in[v]^K} \left(\prod_{r=1}^K \mathbb{P}(\boldsymbol{e}_{i_r})\right)\frac{1}{K}\sum_{r=1}^K \mathrm{LM}_\theta^{(t)}\left(\cdot\mid \boldsymbol{e}_{i_r},\boldsymbol{X}\right) \\
&= \sum_{(i_1,\ldots,i_K)\in[v]^K} \left(\prod_{r=1}^K \alpha_{t,i_r}^{(\mathrm{MTS})}\right)\frac{1}{K}\sum_{r=1}^K \mathrm{LM}_\theta^{(t)}\left(\cdot\mid \boldsymbol{e}_{i_r},\boldsymbol{X}\right) \\
&= \frac{1}{K}\sum_{r=1}^K\sum_{j=1}^v \mathrm{LM}_\theta^{(t)}(\cdot\mid \boldsymbol{e}_j,\boldsymbol{X}) \sum_{(i_1,\ldots,i_{r-1},i_{r+1},\ldots,i_K)\in[v]^{K-1}} \alpha_{t,j}^{(\mathrm{MTS})}\prod_{\substack{s=1\\s\neq r}}^K \alpha_{t,i_s}^{(\mathrm{MTS})} \\
&= \frac{1}{K}\sum_{r=1}^K\sum_{j=1}^v \alpha_{t,j}^{(\mathrm{MTS})}\,\mathrm{LM}_\theta^{(t)}(\cdot\mid \boldsymbol{e}_j,\boldsymbol{X}) \sum_{(i_1,\ldots,i_{r-1},i_{r+1},\ldots,i_K)\in[v]^{K-1}} \prod_{\substack{s=1\\s\neq r}}^K \alpha_{t,i_s}^{(\mathrm{MTS})} \\
&= \sum_{r=1}^K\frac{1}{K}\sum_{j=1}^v \alpha_{t,j}^{(\mathrm{MTS})}\,\mathrm{LM}_\theta^{(t)}\left(\cdot\mid \boldsymbol{e}_j,\boldsymbol{X}\right) = \sum_{j=1}^v \alpha_{t,j}^{(\mathrm{MTS})}\,\mathrm{LM}_\theta^{(t)}\left(\cdot\mid \boldsymbol{e}_j,\boldsymbol{X}\right) \\
&= \sum_{j=1}^v \alpha_{t,j}^{(\mathrm{CoT2})}\,\mathrm{LM}_\theta^{(t)}\left(\cdot\mid \boldsymbol{e}_j,\boldsymbol{X}\right) = \boldsymbol{\alpha}_{t+1}^{(\mathrm{CoT2})}. \quad\quad (8)
\end{aligned}
$$

Thus, combining (7), and (8) completes the induction and our argument:

$$
\boldsymbol{\alpha}_{t+1}^{(\mathrm{disc})} = \mathbb{E}\left[\hat{\boldsymbol{\alpha}}_{t+1}^{(\mathrm{disc})}\right] = \boldsymbol{\alpha}_{t+1}^{(\mathrm{CoT2})} = \mathbb{E}\left[\hat{\boldsymbol{\alpha}}_{t+1}^{(\mathrm{MTS})}\right] = \boldsymbol{\alpha}_{t+1}^{(\mathrm{MTS})}.
$$

$\square$

**Proposition 3.** *(i) Let $\boldsymbol{\alpha}_m$ be the expected output distribution after $m$ steps of CoT according to Proposition 2. Let $\hat{\boldsymbol{\alpha}}_m^{(\mathrm{MTS})}$ be the distribution resulting from averaging the outputs of $N$ i.i.d. CoT2-MTS traces with parallelism $K$. Then, to guarantee $\|\hat{\boldsymbol{\alpha}}_m^{(\mathrm{MTS})} - \boldsymbol{\alpha}_m\|_2 \leq \epsilon$ with high probability, the total number of samples (traces) required scales as $\Theta(K^{-1}\epsilon^{-2})$. (ii) Defining $\hat{\boldsymbol{\alpha}}_m^{(\mathrm{disc})}$ to be the distribution obtained by averaging the outputs of $NK$ i.i.d. discrete CoT traces, we have the following upper bound on the expected error of MTS:*

$$
\mathbb{E}\left[\|\hat{\boldsymbol{\alpha}}_m^{(\mathrm{MTS})} - \boldsymbol{\alpha}_m\|_2^2\right] \leq \left(2 - \frac{1}{K}\right)\mathbb{E}\left[\|\hat{\boldsymbol{\alpha}}_m^{(\mathrm{disc})} - \boldsymbol{\alpha}_m\|_2^2\right].
$$

*Proof.* Our notation in the proof will deviate from that in the proposition statement. As in the proof of the previous proposition, we denote by $\hat{\boldsymbol{\alpha}}_t^{(\mathrm{MTS})}$ a single realization of the distribution obtained from the last $K$ tokens of CoT2-MTS. On the other hand, the random variable $\hat{\boldsymbol{\alpha}}_t^{(\mathrm{disc})}$ represents the distribution obtained by averaging $K$ i.i.d. discrete CoT traces. We will compare the variances of these two approaches. The resulting inequality will similarly hold for the MTS and discrete CoT estimators given in the proposition statement, which use $N$ and $NK$ samples, respectively. Initially, we'll compare the variances of the two approaches at step $t = 2$ (empirical distributions formed at the end of the second sampling step), and a parallel argument will hold for any $t$.

**CoT2-MTS:** In the beginning, we have deterministic distribution $\boldsymbol{\alpha}_1 = \mathrm{LM}_\theta^{(0)}(\cdot \mid \mathbf{X}) = (\alpha_{1,1},\ldots,\alpha_{1,v}) \in \Delta^{v-1}$. At step $t = 1$, we draw $K$ tokens $i_1,\ldots,i_K \overset{\text{i.i.d.}}{\sim} \text{Categorical}(\boldsymbol{\alpha}_1)$. The

empirical frequencies over the vocabulary are:

$$\hat{\alpha}_{1,j}^{(\text{MTS})} = \frac{1}{K} \sum_{r=1}^{K} \mathbf{1}\{i_r = j\}, \qquad j \in [v].$$

This means, the query fed into the model is $z_1 = \sum_{j=1}^{v} \hat{\alpha}_{1,j} \, \mathbf{e}_j$. By Assumption 1, the output distribution is:

$$\hat{\alpha}_2^{(\text{MTS})} = \text{LM}_\theta^{(1)}(\cdot \mid \mathbf{z}_1, X) = \sum_{j=1}^{v} \hat{\alpha}_{1,j}^{(\text{MTS})} \, \text{LM}_\theta^{(1)}(\cdot \mid \mathbf{e}_j, X).$$

Thus $\hat{\alpha}_2$ is a convex combination of the $v$ fixed distributions $\text{LM}_\theta^{(1)}(\cdot \mid \mathbf{e}_j, X)$ for $j \in [v]$. At step 2, we will draw $K$ tokens from this common distribution $\hat{\alpha}_2$. Let

$$Y_r^{(\text{MTS})} \in [v], \quad r = 1, \dots, K,$$
$$Y_r^{(\text{MTS})} \mid \hat{\alpha}_2^{(\text{MTS})} \sim \text{Categorical}(\hat{\alpha}_2^{(\text{MTS})}).$$

be the tokens drawn. Now, we'll calculate the variance along any fixed vocabulary coordinate $k \in [v]$ and define

$$Y_{r,k}^{(\text{MTS})} := \mathbf{1}\{Y_r^{(\text{MTS})} = k\}, \quad \bar{Y}_k^{(\text{MTS})} := \frac{1}{K} \sum_{r=1}^{K} Y_{r,k}^{(\text{MTS})}.$$

Denote the distributions of the drawn tokens:

$$\mathbf{U}_r^{(\text{MTS})} := \text{LM}_\theta^{(1)}(\cdot \mid \mathbf{e}_{i_r}, X) \in \Delta^{v-1},$$
$$U_{r,k}^{(\text{MTS})} := (\mathbf{U}_r^{(\text{MTS})})_k, \quad \bar{U}_k^{(\text{MTS})} := \frac{1}{K} \sum_{r=1}^{K} U_{r,k}^{(\text{MTS})}.$$

Then $U_{r,k}^{(\text{MTS})}$ depends only on the first-step choice $i_r$ and is therefore i.i.d. across $r$. $\bar{U}_k^{(\text{MTS})} = \hat{\alpha}_{2,k}$ is the random success parameter for the Bernoulli variables $Y_{r,k}^{(\text{MTS})}$. By using the unbiasedness given by Proposition 2, we know that $\mathbb{E}[\bar{U}_k^{(\text{MTS})}] = \alpha_{2,k}^{(\text{CoT2})}$. Define the shorthand notations:

$$\mu_k := \mathbb{E}[U_{r,k}^{(\text{MTS})}] = \mathbb{E}[\bar{U}_k^{(\text{MTS})}] = \alpha_{2,k}^{(\text{CoT2})}$$
$$\sigma_k^2 := \text{Var}[U_{r,k}^{(\text{MTS})}].$$

We calculate the variance of the MTS estimator $\text{Var}[\bar{Y}_k^{(\text{MTS})}]$ by using the law of total variance:

$$\text{Var}[\bar{Y}_k^{(\text{MTS})}] = \mathbb{E}\left[\text{Var}[\bar{Y}_k^{(\text{MTS})} \mid \bar{U}_k^{(\text{MTS})}]\right] + \text{Var}\left(\mathbb{E}[\bar{Y}_k^{(\text{MTS})} \mid \bar{U}_k^{(\text{MTS})}]\right).$$

Given $\bar{U}_k^{(\text{MTS})}$, the $Y_{r,k}^{(\text{MTS})}$ are i.i.d. Bernoulli($\bar{U}_k^{(\text{MTS})}$). Thus:

$$\text{Var}\left[\bar{Y}_k^{(\text{MTS})} \mid \bar{U}_k^{(\text{MTS})}\right] = \frac{\bar{U}_k^{(\text{MTS})}(1 - \bar{U}_k^{(\text{MTS})})}{K}.$$

Taking expectation and using $\text{Var}[\bar{U}_k^{(\text{MTS})}] = \sigma_k^2/K$:

$$\mathbb{E}\left[\frac{\bar{U}_k^{(\text{MTS})}(1 - \bar{U}_k^{(\text{MTS})})}{K}\right] = \frac{1}{K}\left(\mathbb{E}[\bar{U}_k^{(\text{MTS})}] - \mathbb{E}[\bar{U}_k^{2,(\text{MTS})}]\right)$$

$$= \frac{1}{K}\left(\mu_k - \left(\sigma_k^2/K + \mu_k^2\right)\right) \tag{A}$$

$$= \frac{\mu_k(1 - \mu_k)}{K} - \frac{\sigma_k^2}{K^2}.$$

Also, the other term is:

$$\text{Var}\left(\mathbb{E}[\bar{Y}_k^{(\text{MTS})} \mid \bar{U}_k^{(\text{MTS})}]\right) = \text{Var}[\bar{U}_k^{(\text{MTS})}] = \frac{\sigma_k^2}{K}. \tag{B}$$

Combining (A) and (B) yields:

$$\text{Var}[\bar{Y}_k^{(\text{MTS})}] = \frac{\mu_k(1 - \mu_k)}{K} + \frac{K - 1}{K^2}\sigma_k^2. \tag{9}$$

**Discrete-CoT:** For $K$ independent traces, each trace $r$ draws at step 2 from its own distribution $\mathbf{U}_r^{(\text{disc})}$ (instead of a common mixture). Let $Y_{r,k}^{(\text{disc})} \sim \text{Bernoulli}(U_{r,k}^{(\text{disc})})$ and $\bar{Y}_k^{(\text{disc})} := \frac{1}{K}\sum_r Y_{r,k}^{(\text{disc})}$. Apply the law of total variance:

$$\text{Var}[Y_{r,k}^{(\text{disc})}] = \underbrace{\mathbb{E}\left[\text{Var}[Y_{r,k}^{(\text{disc})} \mid U_{r,k}^{(\text{disc})}]\right]}_{\text{(A)}} + \underbrace{\text{Var}\left(\mathbb{E}[Y_{r,k}^{(\text{disc})} \mid U_{r,k}^{(\text{disc})}]\right)}_{\text{(B)}}.$$

Given $U_{r,k}^{(\text{MTS})}$, the variance of $Y^{(\text{disc})}$ is:

$$\text{Var}[Y_{r,k}^{(\text{disc})} \mid U_{r,k}^{(\text{disc})}] = U_{r,k}^{(\text{disc})}\left(1 - U_{r,k}^{(\text{disc})}\right).$$

Taking expectation over $U_{r,k}^{(\text{disc})}$,

$$\begin{aligned}
\mathbb{E}\left[U_{r,k}^{(\text{disc})}(1 - U_{r,k}^{(\text{disc})})\right] &= \mathbb{E}[U_{r,k}^{(\text{disc})}] - \mathbb{E}[U_{r,k}^{2,(\text{disc})}] \\
&= \mu_k - \left(\text{Var}[U_{r,k}^{(\text{disc})}] + \mu_k^2\right) \\
&= \mu_k(1 - \mu_k) - \sigma_k^2,
\end{aligned}$$

where $\sigma_k^2 := \text{Var}[U_{r,k}^{(\text{disc})}]$. Note that the variances are the same $\text{Var}[U_{r,k}^{(\text{disc})}] = \text{Var}[U_{r,k}^{(\text{MTS})}] = \sigma_k^2$ since the initial distribution $\alpha_1$ are shared between MTS and discrete CoT approaches. For the conditional-expectation variance term

$$\text{Var}\left(\mathbb{E}[Y_{r,k}^{(\text{disc})} \mid U_{r,k}^{(\text{disc})}]\right) = \text{Var}[U_{r,k}^{(\text{disc})}] = \sigma_k^2.$$

Combining (A) and (B) yields:

$$\text{Var}[Y_{r,k}^{(\text{disc})}] = \left(\mu_k(1 - \mu_k) - \sigma_k^2\right) + \sigma_k^2 = \mu_k(1 - \mu_k).$$

$$\implies \text{Var}[\bar{Y}_k^{(\text{disc})}] = \frac{\mu_k(1 - \mu_k)}{K}. \tag{10}$$

Using (9) and (10), we have:

$$\text{Var}[\bar{Y}_k^{(\text{MTS})}] = \text{Var}[\bar{Y}_k^{(\text{disc})}] + \frac{K - 1}{K^2}\sigma_k^2.$$

$$\implies \frac{\text{Var}[\bar{Y}_k^{(\text{MTS})}]}{\text{Var}[\bar{Y}_k^{(\text{disc})}]} = 1 + \frac{K - 1}{K} \cdot \frac{\sigma_k^2}{\mu_k(1 - \mu_k)}.$$

Because every $U_{r,k}^{(\text{MTS})} \in [0, 1]$, the known inequality $\text{Var}[Z] \leq \mathbb{E}[Z](1 - \mathbb{E}[Z])$ for any random variable $Z \in [0, 1]$ implies $\sigma_k^2 \leq \mu_k(1 - \mu_k)$. As a result:

$$\frac{\text{Var}[\bar{Y}_k^{(\text{MTS})}]}{\text{Var}[\bar{Y}_k^{(\text{disc})}]} \leq 1 + \frac{K - 1}{K} = 2 - \frac{1}{K}.$$

Thus, summing this over all $k \in [v]$ gives:

$$\mathbb{E}\left[\left\|\hat{\alpha}_m^{(\text{MTS})} - \alpha_m\right\|_2^2\right] \leq \left(2 - \tfrac{1}{K}\right)\mathbb{E}\left[\left\|\hat{\alpha}_m^{(\text{disc})} - \alpha_m\right\|_2^2\right].$$

**Proof for any $t$.** We will now show this at any step $t$. Consider the discrete CoT case with K i.i.d. traces. Different from the 2-step case, where the tokens for MTS and discrete CoT at step $t = 1$ are drawn from the same distribution $\alpha_1$, they will be drawn from the empirical distributions $\hat{\alpha}_t^{(\text{MTS})}$ and $\hat{\alpha}_t^{(\text{disc})}$. Similarly define $U_r^{(\text{disc})}$ to be the distribution $\text{LM}_\theta^{(t)}(\cdot \mid \mathbf{e}_{i_r}, X) \in \Delta^{v-1}$ obtained by sampling the token $i_r$. For each vocabulary index $k \in [v]$

$$\sigma_{t,k}^{2,(\text{disc})} := \text{Var}\left[U_{r,k}^{(\text{disc})}\right], \quad \text{Var}\left[\hat{\mu}_{t,k}^{(\text{disc})}\right] = \frac{\sigma_{t,k}^{2,(\text{disc})}}{K}.$$

The equality follows because the $U_{r,k}^{(\text{disc})}$ are i.i.d. across $r$. Now, we calculate the variance at step $t+1$. Each trace $r$ now draws a Bernoulli variable $Y_{r,k}^{(\text{disc})} \sim \text{Bernoulli}(U_{r,k}^{(\text{disc})})$. Parallel to the previous case, using the law of total variance cancels out the variance terms $\sigma_{t,k}^{2,(\text{disc})}$ and yields:

$$\text{Var}\left[\bar{Y}_{t+1,k}^{(\text{disc})}\right] = \frac{\mu_{t+1,k}(1 - \mu_{t+1,k})}{K}. \tag{11}$$

**CoT2-MTS:** This time, all tokens at step $t+1$ are drawn from a shared $\hat{\mu}_t^{(\text{MTS})}$, which is resulted after drawing K tokens from $\hat{\alpha}_t^{\text{MTS}}$. Using $Y_{r,k}^{(\text{MTS})} \mid \hat{\mu}_t^{(\text{MTS})} \sim \text{Bernoulli}(\hat{\mu}_{t,k}^{(\text{MTS})})$ and applying the law of total variance gives:

$$\begin{aligned}
\text{Var}\left[\bar{Y}_{t+1,k}^{(\text{MTS})}\right] &= \mathbb{E}\left[\frac{\hat{\mu}_{t,k}^{(\text{MTS})}(1 - \hat{\mu}_{t,k}^{(\text{MTS})})}{K}\right] + \text{Var}\left[\hat{\mu}_{t,k}^{(\text{MTS})}\right] \\
&= \frac{\mu_{t+1,k}(1 - \mu_{t+1,k})}{K} - \frac{\sigma_{t,k}^{2,(\text{MTS})}}{K^2} + \frac{\sigma_{t,k}^{2,(\text{MTS})}}{K} \\
&= \frac{\mu_{t+1,k}(1 - \mu_{t+1,k})}{K} + \frac{K-1}{K^2}\sigma_{t,k}^{2,(\text{MTS})}.
\end{aligned} \tag{12}$$

Take the ratio of (12) to (11):

$$\frac{\text{Var}[\bar{Y}_{t+1,k}^{(\text{MTS})}]}{\text{Var}[\bar{Y}_{t+1,k}^{(\text{disc})}]} = 1 + \frac{K-1}{K}\frac{\sigma_{t,k}^{2,(\text{MTS})}}{\mu_{t+1,k}(1 - \mu_{t+1,k})}.$$

Again, every $U_{r,k}^{(\text{MTS})}$ is $0 \leq U_{r,k}^{(\text{MTS})} \leq 1$. For any random variable $Z$ bounded in $[0,1]$, we know that $\text{Var}[Z] \leq \mathbb{E}[Z](1 - \mathbb{E}[Z])$. Therefore we always have $\sigma_{t,k}^{2,(\text{MTS})} \leq \mu_{t+1,k}(1 - \mu_{t+1,k})$ and thus, $\text{Var}\left[\hat{\mu}_{t,k}^{(\text{MTS})}\right] \leq \frac{\mu_{t+1,k}(1 - \mu_{t+1,k})}{K}$. Using this in the previous equality yields:

$$\frac{\text{Var}[\bar{Y}_{t+1,k}^{(\text{MTS})}]}{\text{Var}[\bar{Y}_{t+1,k}^{(\text{disc})}]} \leq 1 + \frac{K-1}{K} = 2 - \frac{1}{K}.$$

Thus, defining $\sigma_{\text{disc}}^2 := \mathbb{E}\left[\|\hat{\alpha}_m^{(\text{disc})} - \alpha_m\|_2^2\right]$, we have shown that:

$$\sigma_{\text{disc}}^2 \leq \mathbb{E}\left[\|\hat{\alpha}_m^{(\text{MTS})} - \alpha_m\|_2^2\right] \leq \left(2 - \tfrac{1}{K}\right)\sigma_{\text{disc}}^2. \tag{13}$$

We also know that $\hat{\alpha}_m^{(\text{disc})}$ is the average of K i.i.d. discrete rollouts, and thus:

$$\sigma_{\text{disc}}^2 = \mathbb{E}\left[\|\hat{\alpha}_m^{(\text{disc})} - \alpha_m\|_2^2\right] = \frac{1}{K}\sum_{j=1}^{v}\mu_{m,j}\left(1 - \mu_{m,j}\right) \leq \frac{1}{K}.$$

Define the average of $N$ independent MTS roll-outs as $\bar{\alpha}_m := \frac{1}{N}\sum_{r=1}^{N}\hat{\alpha}_m^{(\text{MTS}),r}$, and because variances add for independent runs, we have $\text{Var}[\bar{\alpha}_m] = \text{Var}[\hat{\alpha}_m^{(\text{MTS})}]/N$. Applying Chebyshev's inequality yields:

$$\mathbb{P}\left[\|\bar{\alpha}_m - \alpha_m\|_2 > \epsilon\right] \leq \frac{(2 - \frac{1}{K})\sigma_{\text{disc}}^2}{N\epsilon^2} \leq \frac{(2 - \frac{1}{K})}{KN\epsilon^2}.$$

In order to make this probability $\leq \delta$, it is enough to take

$$N \geq \frac{2 - \frac{1}{K}}{K\epsilon^2\delta} \implies N = O\left(\tfrac{1}{K\epsilon^2}\right).$$

To finish our argument by finding a lower bound on $N$, we leverage the standard result in multinomial estimation that $\Theta\left(\frac{1}{\epsilon^2}\right)$ i.i.d. samples are necessary and sufficient to learn a $v$-category distribution in $\|\cdot\|_2$-distance $\leq \epsilon$ (Kamath et al., 2015). We know by Proposition 2 that the distribution $\alpha_m^{\text{MTS}}$ to be recovered is shared between MTS and discrete CoT estimators. Additionally, from (13), we

know the variance of the estimator $\hat{\alpha}_m^{\text{(MTS)}}$ is at least as large as that of $\hat{\alpha}_m^{\text{(disc)}}$. Note that the estimator $\hat{\alpha}_m^{\text{(disc)}}$ is the average of $K$ i.i.d. discrete CoT draws, which guarantees an estimation error of at most $\epsilon$ with $\Theta\left(\frac{1}{K\epsilon^2}\right)$ aggregated samples. Hence, the same lower bound applies to MTS samplings, yielding a sample complexity of $N = \Omega\left(\frac{1}{K\epsilon^2}\right)$. Combining with the upper bound, we have $N = \Theta\left(\frac{1}{K\epsilon^2}\right)$ as claimed. This completes the argument.

$\square$

### E.4 Construction for Minimum Non-Negative Sum (MNNS) Task

We describe a single-layer transformer with an attention block followed by a mixture-of-experts (MoE) feed-forward block. Let $n$ be the length of the input sequence of integer tokens. Denote the tokenized input numbers as $z_1, z_2, \ldots, z_n$; and let the arrow ($\rightarrow$) token be denoted as $z_{n+1}$. We also have a dummy input token $z_{n+2}$, which is the embedding corresponding to the number 0, so that we have $n + 2$ tokens initially. We will construct the transformer with $n + 1$ MLPs in the mixture of experts layer, where the first $n$ are partial-sum MLPs and the last one is the MLP that reads off the answer from among all the stored partial sums after $m$ steps. We start with the following assumption on the structure of the tokens.

**Remark 2.** *As an empirical validation of Proposition 1, we observe that training according to this construction with trigonometric embeddings yields perfect accuracy.*

**Assumption 2.** *Let $d = d_e + d_p$ be the embedding size where $d_e = 2^{n+1}$ and $d_p = n + 2$. The token embeddings are on the first $d_e$ coordinates, while the positional encodings are on the last $d_p$ coordinates and are one-hot encoded. where each $z_i = \begin{pmatrix} e_i \\ \mathbf{p}_i \end{pmatrix} \in \mathbb{R}^{d_e + d_p}$ is formed by vertically concatenating a content embedding $e_i \in \mathbb{R}^{d_e}$ and a positional encoding $\mathbf{p}_i \in \mathbb{R}^{d_p}$. We assume each $\mathbf{p}_i$ is a one-hot vector in $\mathbb{R}^{d_p}$, so that $\mathbf{p}_i^\top \mathbf{p}_j = 0$ for $i \neq j$, and $\|\mathbf{p}_i\| = 1$.*

We now state the following proposition, which helps us to attend and select the input tokens $z_1, \ldots, z_{n+1}$ one by one by the attention block.

**Proposition 4.** *Suppose we have $n + 2$ tokens $\{z_1, z_2, \ldots, z_{n+2}\}$ in $\mathbb{R}^d$, each of the form $z_i = \begin{pmatrix} e_i \\ p_i \end{pmatrix}$, where $e_i \in \mathbb{R}^{d_e}, p_i \in \mathbb{R}^{d_p}, d = d_e + d_p$. Let $p_1, p_2, \ldots, p_{n+2} \in \mathbb{R}^{d_p}$ be orthonormal set of positional vectors according to Assumption 2. Then, there exists a rotation matrix $R \in \mathbb{R}^{d_p \times d_p}$ satisfying $Rp_j = p_{j-1 \bmod (n+2)}$ for all $j \in [n + 2]$, and the block matrices*

$$W = \begin{pmatrix} \mathbf{0}_{d_e \times d_e} & \mathbf{0}_{d_e \times d_p} \\ \mathbf{0}_{d_p \times d_e} & c \cdot R \end{pmatrix} \in \mathbb{R}^{d \times d} \quad and \quad W_v = \begin{pmatrix} I_{d_e} & \mathbf{0}_{d_e \times d_p} \\ \mathbf{0}_{d_p \times d_e} & I_{d_p} \end{pmatrix} \in \mathbb{R}^{d \times d}$$

*with $c \rightarrow \infty$, ensure that the attention block*

$$\text{Attn}(z, Z) = \mathbb{S}\left(z^\top W Z^\top\right) Z W_v,$$

*performs a cyclic next-index selection: if the query is $z_i$, it selects column $j^* \equiv (i + 1)$ (mod $n + 2$) from $Z$ and returns $z_{j^*}$.*

*Proof. Definition of Matrix $W$.* We will first construct a rotation matrix. We have $n + 2$ orthonormal position vectors $p_1, \ldots, p_{n+2} \in \mathbb{R}^{d_p}$. Then, $R$ is the following $(n + 2) \times (n + 2)$ permutation matrix

$$R = \begin{pmatrix} 0 & 1 & 0 & \cdots & 0 & 0 \\ 0 & 0 & 1 & \cdots & 0 & 0 \\ 0 & 0 & 0 & \cdots & 0 & 0 \\ \vdots & \vdots & \vdots & \ddots & \vdots & \vdots \\ 0 & 0 & 0 & \cdots & 0 & 1 \\ 1 & 0 & 0 & \cdots & 0 & 0 \end{pmatrix},$$

which cyclically shifts the basis vectors $p_j$ backward by one index, i.e., $Rp_j = p_{j-1 \bmod (n+2)}$. Then, we specify

$$W = \begin{pmatrix} \mathbf{0}_{d_e \times d_e} & \mathbf{0}_{d_e \times d_p} \\ \mathbf{0}_{d_p \times d_e} & c \cdot R \end{pmatrix} \in \mathbb{R}^{d \times d}.$$

Hence for $z_i = (e_i; p_i)$, we have $\left(e_i^\top, p_i^\top\right) W = \left(0, p_i^\top R\right)$. Thus the dot-product with $z_j$ is

$$\left(0 \ \ p_i^\top R\right)\begin{pmatrix} e_j \\ p_j \end{pmatrix} = p_i^\top R p_j.$$

Since positional encodings are orthogonal, we know that:

$$p_i^\top R p_j = \begin{cases} 1, & j \equiv i + 1(\mathrm{mod}(n+2)), \\ 0, & \text{else.} \end{cases}$$

So row-wise softmax $\mathbb{S}\left(x^\top W X^\top\right)$ places all probability mass at column $j^* \equiv i + 1(\mathrm{mod}(n+2))$ by saturating softmax at position $j$ as $c \to \infty$.

*Definition of Matrix $W_v$.* In this case, we simply set $W_v = I_d$, and thus, once the row-wise softmax selects column $j^*$ with probability 1, we have

$$z_{j^*}^\top W_v = z_{j^*},$$

so the final output is precisely the chosen $z_{j^*}$. This completes the construction. $\qquad\square$

Having defined the attention block, we state the following proposition that helps selecting different MLPs for the tokens $z_1, \ldots, z_{n+1}$ outputted by the attention block.

Having defined the attention block, we now show how a mixture-of-experts layer can exclusively select MLP$_i$ for each token $z_i$, $i = 1, \ldots, n + 1$ outputted by the attention block.

**Proposition 5.** *Let $MLP_1, \ldots, MLP_{n+1}$ be $n + 1$ experts in a mixture-of-experts (MoE) module. Suppose we have $n + 1$ fixed token embeddings $\{z_1, z_2, \ldots, z_{n+1}\} \subset \mathbb{R}^d$, where each token is formed according to Assumption 2. Given routing parameters $W = [w_1 \ \ldots \ w_{n+1}]^\top$, define the MoE feed-forward block as*

$$\mathrm{MoEBlock}(z) = \sum_{j=1}^{n+1} \left[ \mathrm{Softmax}(Wz)_j \cdot MLP_j(z) \right],$$

*where*

$$\mathrm{Softmax}(Wz)_j = \frac{\exp\left(w_j^\top z\right)}{\sum_{k=1}^{n+1} \exp\left(w_k^\top z\right)}, \quad j = 1, \ldots, n + 1.$$

*There exist routing matrix $W \in \mathbb{R}^{(n+1)\times d}$ such that the distribution $\mathrm{Softmax}(c \cdot Wz_i)$ as $c \to \infty$ assigns a weight of 1 on $MLP_i$ when $z_i$ is given as input.*

*Proof.* We partition $w_j$ to ignore the content embedding $e_i$ and match the positional block $p_j$. Concretely, write $w_j = \begin{pmatrix} 0_{d_e} \\ p_j \end{pmatrix}$. Then, for each token $z_i = (e_i; p_i)$,

$$w_j^\top z_i = \left(0_{d_e}^\top \ \ p_j^\top\right)\begin{pmatrix} e_i \\ p_i \end{pmatrix} = p_j^\top p_i.$$

Since $p_j^\top p_i = \delta_{ij}$, we have $w_j^\top z_i = \delta_{ij}$. Therefore, the softmax evaluates to

$$\lim_{c\to\infty} \mathrm{Softmax}(c \cdot Wz_i)_j \to \frac{\exp(c \cdot \delta_{ij})}{\sum_{k=1}^{n+1} \exp(c \cdot \delta_{ik})} = \delta_{ij}.$$

In other words, $\mathrm{Softmax}(c \cdot Wz_i)$ places all mass on expert $j = i$. Thus each token $z_i$ (for $i = 1, \ldots, n+1$) deterministically selects the $i$-th expert MLP$_i$. $\qquad\square$

In the next proposition, we show how to iteratively expand the partial sums by adding and subtracting the digit obtained from the attention block and write each resulting sum to a distinct spot in the output vector.

**Proposition 6** (Partial-Sum MLPs). *Suppose that the embedding dimension $d$ satisfies $d \geq 2^{j+1} + d_p$. Let $z_{prev}$ contain the $2^{j-1}$ partial sums $s_k$ each encoded by a pair $(\cos(\omega s_k),\ \sin(\omega s_k))$ of coordinates such that:*

$$z_{prev} = \left[\cos(\omega s_1)\ \sin(\omega s_1)\ \ldots\ \cos(\omega s_{2^{j-1}})\ \sin(\omega s_{2^{j-1}})\, 0 \ldots 0\right]^\top \in \mathbb{R}^d,$$

*and let $z_{curr}$ contain the input digit $d_j$ encoded in the first two coordinates:*

$$z_{curr} = \left[\cos(\omega d_j)\ \sin(\omega d_j)\, 0\ \ldots\ 0\right]^\top \in \mathbb{R}^d.$$

*Then, for any $1 \leq j \leq n$, there exist $MLP_j : \mathbb{R}^d \times \mathbb{R}^d \to \mathbb{R}^d$ such that when $(z_{prev}, z_{curr})$ is given as input, it outputs the vector $z_{out} \in \mathbb{R}^d$ so that its first $2^j$ coordinate-pairs store the trigonometric encodings of $(s_k + d_j)$, and the next $2^j$ coordinate-pairs store those of $(s_k - d_j)$. Formally, first $2^j$ coordinates are $[\cos(\omega(s_k + d_j)), \sin(\omega(s_k + d_j))]$ for all partial sums $s_k$, and the next $2^j$ coordinates are $[\cos(\omega(s_k - d_j)), \sin(\omega(s_k - d_j))]$ for all partial sums $s_k$, with any remaining coordinates set to zero.*

*Proof.* Each expert $MLP_j$ (for $1 \leq j \leq n$) adds $j$-th integer $d_j$ in both its positive and negative form to all previously computed partial sums. For simplicity, let's say that $j$-th integer to add is $d_j$. By trigonometric identities, we know that

$$\cos(\omega(s_k + d_j)) = \cos(\omega s_k)\cos(\omega d_j) - \sin(\omega s_k)\sin(\omega d_j),$$
$$\sin(\omega(s_k + d_j)) = \sin(\omega s_k)\cos(\omega d_j) + \cos(\omega s_k)\sin(\omega d_j),$$

and similarly,

$$\cos(\omega(s_k - d_j)) = \cos(\omega s_k)\cos(\omega d_j) + \sin(\omega s_k)\sin(\omega d_j),$$
$$\sin(\omega(s_k - d_j)) = \sin(\omega s_k)\cos(\omega d_j) - \cos(\omega s_k)\sin(\omega d_j).$$

Using the above identities, we will obtain the sum by introducing matrices that do shift/swap operations. Concretely, for $k = 1, \ldots, 2^m$, the $k$-th $2 \times 2$ block acts on $\begin{pmatrix}\cos(\omega s_k) \\ \sin(\omega s_k)\end{pmatrix}$ in $z_{prev}$. We define:

$$W_{\sin}^{+} = \mathrm{diag}\left(\underbrace{\begin{pmatrix}0 & -1 \\ 1 & 0\end{pmatrix}, \ldots, \begin{pmatrix}0 & -1 \\ 1 & 0\end{pmatrix}}_{2^{j-1}\ \text{blocks}}, 0, \ldots, 0\right),$$

$$W_{\sin}^{-} = \mathrm{diag}\left(\underbrace{\begin{pmatrix}0 & 1 \\ -1 & 0\end{pmatrix}, \ldots, \begin{pmatrix}0 & 1 \\ -1 & 0\end{pmatrix}}_{2^{j-1}\ \text{blocks}}, 0, \ldots, 0\right).$$

The above constructions of $W_{\sin}^{+}$ and $W_{\sin}^{-}$ satisfy,

$$W_{\sin}^{+} z_{prev} = \left[-\sin(\omega s_1)\ \cos(\omega s_1) \cdots -\sin(\omega s_{2^{j-1}})\ \cos(\omega s_{2^{j-1}})\, 0 \ldots 0\right]^\top \in \mathbb{R}^d$$

and

$$W_{\sin}^{-} z_{prev} = \left[\sin(\omega s_1)\ -\cos(\omega s_1) \ldots \sin(\omega s_{2^{j-1}})\ -\cos(\omega s_{2^{j-1}})\, 0 \ldots 0\right]^\top \in \mathbb{R}^d.$$

Each of these acts blockwise on the first $2^j$ coordinates of $z_{prev}$ and zeroes out everything else in dimension $d$. We also have $z_{curr} \in \mathbb{R}^d$ with two designated coordinates $z_{curr,1} = \cos(\omega d_j)$, and $z_{curr,2} = \sin(\omega d_j)$, with all other coordinates being zero. We multiply $z_{prev}$ by $\cos(\omega d_j)$ and $\sin(\omega d_j)$ elementwise. Formally, the sum

$$z_{curr,1} \cdot z_{prev} + z_{curr,2} \cdot (W_{\sin}^{+} z_{prev})$$

gives the $2^{j-1}$ partial sums $\{s_k + d_j\}_{k=1}^{2^{j-1}}$ stored in the coordinates from 1 to $2^j$. We define $W_{shift} \in \mathbb{R}^{d \times d}$ in a block form with three row blocks and two column blocks:

$$W_{shift} = \begin{pmatrix} \mathbf{0}_{2^j \times 2^j} & \mathbf{0}_{2^j \times (d-2^j)} \\ I_{2^j} & \mathbf{0}_{2^j \times (d-2^j)} \\ \mathbf{0}_{(d-2^{j+1}) \times 2^j} & \mathbf{0}_{(d-2^{j+1}) \times (d-2^j)} \end{pmatrix}.$$

When applied, the above matrix shifts the first $2^j$ entries of $z_{\text{prev}}$ by $2^j$ coordinates. Now, also define

$$z_{\text{curr},2} \cdot \left( W_{\text{shift}} W_{\sin}^- z_{\text{prev}} \right) + z_{\text{curr},1} \cdot \left( W_{\text{shift}} z_{\text{prev}} \right).$$

This way, the above sum gives us the $2^{j-1}$ partial sums $\{s_k - d_j\}_{k=1}^{2^{j-1}}$ stored in the coordinates from $2^j + 1$ to $2^{j+1}$ encoded in trigonometric format. Then, we normalize this output of the model by $1/2$ and obtain the following output:

$$\left( z_{\text{curr},1} \cdot z_{\text{old}} + z_{\text{curr},2} \cdot \left( M_{\sin}^+ z_{\text{old}} \right) + z_{\text{curr},2} \cdot \left( W_{\text{shift}} W_{\sin}^- z_{\text{prev}} \right) + z_{\text{curr},1} \cdot \left( W_{\text{shift}} z_{\text{prev}} \right) \right)$$
$$= \Big[ \cos\left( \omega \left( s_1 + d_j \right) \right), \sin\left( \omega \left( s_1 + d_j \right) \right), \dots, \cos\left( \omega \left( s_{2^{j-1}} + d_j \right) \right), \sin\left( \omega \left( s_{2^{j-1}} + d_j \right) \right),$$
$$\cos\left( \omega \left( s_1 - d_j \right) \right), \sin\left( \omega \left( s_1 - d_j \right) \right), \dots, \cos\left( \omega \left( s_{2^{j-1}} - d_j \right) \right), \sin\left( \omega \left( s_{2^{j-1}} - d_j \right) \right),$$
$$0, \dots, 0]^\top \in \mathbb{R}^d.$$

Thus, this is exactly the representation of $2^j$ partial sums. This completes the argument. We should remark that, the above argument utilizes a *gated MLP* which explicitly multiplies the elements of the input features, namely, $z_{\text{curr}}$ with the partial sums $z_{\text{prev}}$. On the other hand, we don't require any nonlinear activation function, so our MLP constructions have the form $\text{MLP}(z) = W_3(W_1 z \odot W_2 z)$ for suitable choices of $W_1, W_2, W_3$ where $\odot$ denotes the Hadamard product. The use of gated MLPs is a standard practice in transformer architectures (Shazeer, 2020). □

**Proposition 7** (Read-Off MLP). *Suppose that every partial sum $s_k$ is in the range $[-S, S]$ and let $\omega < \pi/2S$. Assume that the vector*

$$z = [\cos(\omega s_1), \ \sin(\omega s_1), \ \dots, \ \cos(\omega s_{2^n}), \ \sin(\omega s_{2^n}), 0, \dots, 0]^\top \in \mathbb{R}^d,$$

*contains $2^n$ partial sums $\{s_1, \dots, s_{2^n}\}$ encoded in trigonometric form, where $d = 2^{n+1} + n + 2$. Then there exists a single feed-forward network $\text{MLP}_{n+1} : \mathbb{R}^d \to \mathbb{R}^d$ such that, given input $z$, it selects the smallest nonnegative $s_\ell$ from $\{s_1, \dots, s_{2^n}\}$ and outputs the embedding $e_{s_\ell} \in \mathbb{R}^d$, where $s_\ell$ is that minimal nonnegative partial sum.*

**Remark:** Our construction relies on gated MLP, rather than standard MLP, as in Proposition 6.

*Proof.* We know that the input embedding $z$ represents $2^n$ pairs, each pair $(\cos(\omega s_i), \sin(\omega s_i))$ stored consecutively. That is,

$$z = [\cos(\omega s_1), \ \sin(\omega s_1), \ \dots, \ \cos(\omega s_{2^n}), \ \sin(\omega s_{2^n}), 0, \dots, 0]^\top \in \mathbb{R}^d,$$

We will identify the smallest $s_\ell \geq 0$ and output an embedding $e_{s_\ell}$ denoting that integer. We are given that $\omega$ is small enough such that when $s_\ell \in [0, S]$, we ensure $S\omega < \pi/2$. This guarantees $\sin(\omega s_\ell) \geq 0$ if and only if $s_\ell \geq 0$. First, we wish to collapse $z$ into a single vector of size $2^n$, keeping $\cos(\omega s_\ell)$ only when $\sin(\omega s_\ell) \geq 0$ and zeroing it out otherwise. We define two matrices $W_{\cos}, W_{\sin} \in \mathbb{R}^{d \times d}$ by

$$(W_{\cos})_{i,(2i-1)} = 1, \quad (W_{\cos})_{i,j} = 0 \quad \text{for } j \neq 2i - 1,$$
$$(W_{\sin})_{i,(2i)} = 1, \quad (W_{\sin})_{i,j} = 0 \quad \text{for } j \neq 2i.$$

for $1 \leq i \leq 2^n$ and all other rows/columns of $W_{\sin}, W_{\cos}$ are zero. Hence each matrix picks out alternate coordinates:

$$z_{\cos} = W_{\cos} z = \begin{bmatrix} \cos(\omega s_1) \\ \cos(\omega s_2) \\ \vdots \\ \cos(\omega s_{2^n}) \\ 0 \\ \vdots \\ 0 \end{bmatrix} \in \mathbb{R}^d, \quad z_{\sin} = W_{\sin} z = \begin{bmatrix} \sin(\omega s_1) \\ \sin(\omega s_2) \\ \vdots \\ \sin(\omega s_{2^n}) \\ 0 \\ \vdots \\ 0 \end{bmatrix} \in \mathbb{R}^d.$$

In order to find the minimum non-negative number, we need to find the number $s$ such that it maximizes $\cos(\omega s)$ and satisfies $\sin(\omega s) \geq 0$. For this, we utilize a sigmoid activation function in the following way:

$$z_{\text{filter}} = z_{\cos} \odot \sigma\left( c \, z_{\sin} \right),$$

where $\sigma(x) = \frac{1}{1+\exp(-x)}$ is element-wise sigmoid function, and $c \to \infty$ is a large constant. With this choice of $c$, the sigmoid output will be 1 when $s_\ell \geq 0$ and 0 otherwise. Therefore, the resulting vector $z_{\text{filter}}$ contains $\cos(\omega s)$ values at indices where $\sin(\omega s)$ is positive. Now, for $0 \leq s_\ell \leq S$ with $S\omega \leq \frac{\pi}{2}$, the ordering of $s_\ell$ from smallest to largest is the same as the ordering of $\cos(\omega s_\ell)$ from largest to smallest. Thus, to find the minimum nonnegative sum, we find the partial sum $\ell^*$ that maximizes $\cos(\omega s_\ell)$. Utilizing another gating, we calculate

$$\text{Softmax}\left(c\, z_{\text{filter}}\right)^\top z_{\text{filter}}$$

as $c \to \infty$. The softmax vector will be one-hot with 1 at index $\ell^*$ that has the largest $\cos(\omega s_\ell)$. A second multiplication with $z_{\text{filter}}$ will return this $\cos(\omega s_{\ell^*})$. Therefore, $\text{Softmax}\left(c\, z_{\text{filter}}\right)^\top z_{\text{filter}} = \cos(\omega s_{\ell^*})$. Next, we retrieve the corresponding sine entry of $s_{\ell^*}$ by applying the same one-hot selection to $z_{\text{sin}}$. Formally,

$$\text{Softmax}\left(c\, z_{\text{filter}}\right)^\top z_{\text{sin}} = \sin\left(\omega s_{\ell^*}\right),$$

as $c \to \infty$. Hence, from these two selected coordinates, $[\cos(\omega s_{\ell^*}),\ \sin(\omega s_{\ell^*})]$, we produce the final embedding in $\mathbb{R}^d$ by placing them in the first two coordinates and zeros elsewhere:

$$\boldsymbol{e}_{s_{\ell^*}} = [\cos(\omega s_{\ell^*}),\ \sin(\omega s_{\ell^*}),\ 0,\ldots,0]^\top,$$

where $s_{\ell^*}$ is the minimal nonnegative sum. This completes the argument. $\qquad\square$

**Proposition 1** (Solving MNNS). *There exists a 1-layer transformer architecture with embedding dimension $d = d_e + d_p$, where $d_e = 2^{n+1}$ is the state-encoding dimension and $d_p = n + 2$ is the positional-encoding dimension, that solves the MNNS task using CoT2 by storing sine–cosine embeddings of all $2^k$ states at the k-th iteration in a non-overlapping manner.*

*Proof.* We will argue that by combining Propositions 5 to 7, we obtain a single-layer transformer that is formed by an attention block followed by an MoE feed-forward block, which solves the Minimum Non-Negative Sum (MNNS) task.

Suppose that we have $n$ input integers $d_1, \ldots, d_n$, encoded as $z_1, \ldots, z_n$, plus an arrow ($\to$) token $z_{n+1}$ and a dummy token $z_{n+2}$ corresponding to the integer 0. In this case, we will output the tokens representing the ground-truth sums $s_1, \ldots, s_n$, therefore, the number of output tokens is $m = n$ in the MNNS setting. We assume that the inputs are encoded according to Assumption 2. By Proposition 6, there exist $\text{MLP}_1, \ldots, \text{MLP}_n$ that perform the following: whenever $\text{MLP}_j$ is selected with input $(z_{\text{prev}}, z_{\text{curr}})$ such that $z_{\text{prev}}$ stores $2^{j-1}$ partial sums and $z_{\text{curr}}$ stores the digit $d_j$, it adds and subtracts $d_j$ to all previously stored partial sums and stores the resulting $2^j$ partial sums in $z_{\text{out}}$. The dummy token $z_{n+2}$ that corresponds to the integer 0 allows us to initialize the partial sums from zero. If the query token is $z_{n+2}$, we produce the first partial sums by combining this dummy 0 with $d_1$, which are $(+d_1)$ and $(-d_1)$ encoded in an output token.

We assign positional encodings cyclically to output tokens. That means, the first $n + 2$ input tokens have positional encodings from $\boldsymbol{p}_1$ to $\boldsymbol{p}_{n+2}$, and the output tokens have $\boldsymbol{p}_1, \boldsymbol{p}_2, \ldots,$ as their positional encodings, in this exact order. This way, by Proposition 4, $\text{Attn}(z, \mathbf{Z})$ attends and selects the input digit tokens $z_1, z_2, \ldots, z_n$ and finally arrow $z_{n+1}$ one by one and feeds to $\text{MoEBlock}(\cdot)$.

By Proposition 5, there's a $\text{MoEBlock}(z)$ such that if the input is $z_j$ (for $j \leq n$), $\text{MLP}_j$ is selected with probability 1, and if the input is arrow token $z_{n+1}$, $\text{MLP}_{n+1}$ is selected with probability 1, which is the MLP to read-off the final answer. In the input tokens $z_1, \ldots, z_n$, the first two coordinates store the trigonometric representation of $d_1, \ldots, d_n$. To allow outputting the final answer by $\text{MLP}_{n+1}$, the partial sums obtained in the intermediate steps need to be written to separate coordinates. Therefore, $\text{MLP}_j$ takes a vector filled in the first $2^j$ coordinates, adds $d_j$ and writes to the first $2^j$ coordinates, subtracts $d_j$ and writes to the next $2^j$ coordinates, and finally divides the entire representation by 2 to maintain consistent scaling since the number of partial sums is doubled. In other words, the first $n$ MLPs have some repeated behavior. Finally, by Proposition 7, $\text{MLP}_{n+1}$ receives a vector that encodes all $2^n$ possible partial sums in cos/sin form in $2^{n+1}$ coordinates and extracts the embedding of the smallest nonnegative number among them.

Altogether, this single-layer transformer with an attention module to pass the tokens to the mixture-of-experts MLP solves the Minimum Non-Negative Sum task by following CSFT described in 3. $\qquad\square$

