# OpenReview forum: "Continuous Chain of Thought Enables Parallel Exploration and Reasoning"
_ICLR.cc/2026/Conference — ICLR 2026 Poster_

### Official Review · Reviewer_EUvY · 2025-10-20

**Soundness:** 2
**Presentation:** 3
**Contribution:** 2
**Rating:** 4
**Confidence:** 4

**Summary:**

This paper introduces Continuous Chain of Thought (CoT2), a novel latent chain-of-thought method in which the model outputs a convex combination of vocabulary embeddings. This allows the model to explore multiple reasoning paths in parallel within a single forward pass. The authors also propose a method for supervised training, called Continuous Supervised Fine-Tuning (CSFT), and a GRPO based reinforcement learning objective for policy refinement. Empirically, they show that CoT2 outperforms prior approaches such as COCONUT, discrete CoT, and no-CoT on the MNNS, ProsQA, and ProntoQA benchmarks, while converging faster. The experiments also show that the embedding dimension determines the model’s capacity for parallel reasoning, and that reinforcement learning further boosts accuracy and confidence.

**Strengths:**

1. The paper is well-motivated and introduces a clear novelty over previous methods. CoT2 itself, as well as the supervised training and reinforcement learning methods are original.

2. The paper provides a theoretical analysis of CoT2, including a proof showing that a single-layer transformer can solve the MNNS problem, and bounds demonstrating improved sample efficiency compared to discrete CoT.

3. The theoretical claims are backed by empirical results showing that CoT2 outperforms prior approaches such as COCONUT, discrete CoT, and no-CoT on the MNNS, ProsQA, and ProntoQA benchmarks, while converging faster. Additionally, the experiments show that the prosed RL-based method further boosts the accuracy.

**Weaknesses:**

1. A key limitation of CSFT is the reliance of some ground-truth distribution over all intermediate reasoning steps. This is only feasible for the toy tasks like the ones considered in this paper. This dependence seems to hinder the method's applicability to more open-ended reasoning tasks, where such an oracle not available.

2. The need for a new task like the MNNS is not discussed. It seems like the existing datasets are perfectly suited for the methods introduced in the paper, therefore in the current form it seems like its main purpose is to be used in the proof from Section 4, showing that it can be solved by a transformer.

3. Computing each dense vector involves performing a matrix-vector multiplication, where the matrix contains the embeddings of all tokens in the vocabulary. It seems like for large vocabularies, this could be a bottleneck that would reduce the practical gains from parallel explorations of reasoning trajectories.

**Questions:**

See the comments from the weaknesses section. Additionally, have you tested CoT2 on other benchmarks such as GSM8K? What would training look like in this setting? It would be quite interesting to see how it compares to COCONUT, since the authors observed a performance decrease compared to discrete CoT.

---

> ### Author Response · Authors · 2025-11-22
>
> We thank the reviewer for recognizing the significance of our work, and the comprehensiveness of our results, as well as for the detailed comments. Below, we address all of the points raised.
>
> > Weakness 1: A key limitation of CSFT is the reliance of some ground-truth distribution over all intermediate reasoning steps. This is only feasible for the toy tasks like the ones considered in this paper. This dependence seems to hinder the method's applicability to more open-ended reasoning tasks, where such an oracle not available.
>
> **Response:** Thank you for bringing this up. We would like to clarify that CSFT is designed as a **generic supervision framework** as it specifies how to turn a set of trajectories into token‑level targets, but leaves what trajectories to use to the task at hand. Importantly, CSFT does **not** require a full oracle distribution over all intermediate steps. Instead, it uses the supervision obtained by normalizing the distribution over the intermediate states visited at step $t$ by a **selected set of trajectories**. These trajectories may come from a simple search over the underlying graph as in MNNS, Pronto/ProsQA tasks, possibly with Top-$B$ trajectory filtering. The budget $B$ controls how many trajectories we ask the model to track, from standard discrete CoT when $B = 1$ to full superposition when $B = |\mathcal{T}|$. For example, if full budget $B=| \mathcal{T} |$, that means we're simply tracking all possible reasoning paths **without needing the ground-truth intermediate steps**, and the **only requirement is to have the final answer** to supervise the last step.
>
> For more open‑ended reasoning tasks, trajectories for CSFT can be generated by a stronger teacher LLM using beam search or best‑first search over CoT rollouts, then optionally filtered to Top-$B$ trajectories by a verifier or heuristic and converted into $\alpha_t^*$ by normalizing over the states they visit. We also note that one can skip CSFT entirely and apply GRPO on top of a standard SFT model, using only a sparse final‑answer reward through MTS rollouts, this also allows the model to learn to generate continuous tokens without any explicit intermediate supervision.
>
>
> > Weakness 2: The need for a new task like the MNNS is not discussed. It seems like the existing datasets are perfectly suited for the methods introduced in the paper, therefore in the current form it seems like its main purpose is to be used in the proof from Section 4, showing that it can be solved by a transformer.
>
> **Response:** We thank the reviewer for the valuable question. We agree that ProsQA and ProntoQA tasks require multi‑hop reasoning and are well suited to our method, and the reviewer is correct that the one-layer MNNS construction that we discovered through experiments gives us useful insight on how CoT2 packs more information than discrete CoT to solve this task more efficiently. However, beyond inspiring for theoretical results, the main purpose of introducing MNNS as a benchmark is to enrich the evaluation as it's **more structured and mechanistic** task with clear verification, which allows us to demonstrate CoT2's parallel search capability in a rigorous way. MNNS generalizes classical subset‑sum/partition problem and we initially proposed it as the **analytically tractable analogue of the Countdown** game, which is a widely used benchmark for LLM reasoning and RL methods [1, 2]. Concretely, **different than the Countdown game**, MNNS restricts the model to only two deterministic choices (adding or subtracting the next integer) at each step, so the state is a running sum with a known transition graph. This yields a clean search space where we can enumerate all trajectories, compute exact intermediate supervision $\alpha_t^*$, and directly analyze how CoT2 represents many candidate paths within a single continuous token.
>
> [1] Kanishk Gandhi, Ayush Chakravarthy, Anikait Singh, Nathan Lile, and Noah D. Goodman. *Cognitive behaviors that enable self-improving reasoners, or, four habits of highly effective stars*. arXiv:2503.01307, 2025.
>
> [2] Tian Qin, David Alvarez-Melis, Samy Jelassi, and Eran Malach. *To Backtrack or Not to Backtrack: When Sequential Search Limits Model Reasoning*. arXiv:2504.07052, 2025.

---

> > ### Author Response · Authors · 2025-11-22
> >
> > > Weakness 3:  Computing each dense vector involves performing a matrix-vector multiplication, where the matrix contains the embeddings of all tokens in the vocabulary. It seems like for large vocabularies, this could be a bottleneck that would reduce the practical gains from parallel explorations of reasoning trajectories.
> >
> > Response: Thank you for pointing this out. Forming the CoT2 output $z_t = E^\top\alpha_t$ has a cost of $O(vd)$. On the other hand, every decoding step in a standard LM already pays an $O(vd)$ matrix–vector multiplication cost to obtain the final logits $\ell_t = W_{\text{out}} h_t \in \mathbb{R}^{v}$ where $\alpha_t = \mathrm{softmax} (\ell_t)$ with the unembedding matrix $W_{\text{out}}\in\mathbb{R}^{v\times d}$. Thus, **CoT2 adds one more multiply of the same cost**. Meanwhile, this extra $O(vd)$ computation is still **order-wise cheaper** than the attention cost, and **it does not grow with the context length $N$**. For each new token, the per‑layer attention cost is $O(N^2 d)$ and the MLP cost is $O(d^2)$, and across $L$ layers, this yields a total cost of $O(L(N^2d + d^2))$. As a concrete example, the Qwen2.5‑72B model has specifications $d=8192$, $L=80$, and vocabulary size $\approx 150k$. Under realistic scenarios with this model, we typically have $v < L(N^2 + d)$ and this **extra multiplication cost is minor**. Note that, even in the case of full KV cache, the per token inference cost of the transformer stack is $O(L(Nd + d^2))$, so the comparison remains the same as we still have $v < L(N+d)$. Moreover, as we observe in Fig. 2a, discrete CoT needs multiple i.i.d. rollouts (Pass@k) to match CoT2’s single‑shot accuracy, which further supports the point that **CoT2 does not cause a practical computational bottleneck**.
> >
> > > Q1. Additionally, have you tested CoT2 on other benchmarks such as GSM8K? What would training look like in this setting? It would be quite interesting to see how it compares to COCONUT, since the authors observed a performance decrease compared to discrete CoT.
> >
> > **Response:** We thank the reviewer for this question. We would like to highlight that CoT2’s continuous tokens are simplex‑weighted combinations of vocabulary tokens, and our CSFT supervision **explicitly targets implicit parallelism** by superposing thought trajectories with a controllable budget $B$. This makes our method particularly well-suited for enabling parallel reasoning. Because of this, we evaluated our method mainly on **search-heavy tasks** like ProsQA, ProntoQA, and MNNS, and we did not include GSM8K in this submission. However, we agree that evaluating CoT2 on GSM8K would be interesting, and we plan to try using GRPO with CoT2-MTS under curriculum-learning schedule and sparse reward scenario, starting from an SFT model, during this rebuttal period if time permits.

---

> > > ### Comment · Reviewer_EUvY · 2025-11-25
> > >
> > > I thank the authors for the detailed responses; these answer most of my concerns. I would be interested in seeing the new results, if time allows.

---

### Official Review · Reviewer_rbV9 · 2025-10-26

**Soundness:** 3
**Presentation:** 3
**Contribution:** 4
**Rating:** 8
**Confidence:** 3

**Summary:**

The authors propose an approach for continuous chain of thought, where at each step, the model stores a weighted average of all token embeddings weighted by their logit weights. They argue that this allows models to explore several possible reasoning traces in parallel. To perform this technique in practice, the authors propose a supervision approach based on selecting a number of high-performing trajectories in search problems. They show that this approach performs better than standard CoT in the minimum non-negative sum (MNNS) task. They also show theoretically how models with CoT2 are capable of solving this problem. The authors conclude by adapting GRPO for CoT2, which improves accuracy on MNNS.

**Strengths:**

1. The approach is natural and intuitive, and backed up by experiments and theory
2. The presentation is fairly clear overall
3. The topic is very relevant for the conference and of high impact

**Weaknesses:**

1. Some points could be described more clearly (see questions)

### Overall evaluation
This seems like a strong paper with a good blend of theory and experiments, proposing a continuous CoT approach that is reasonable and seems to perform well. I'm not familiar enough with the continuous CoT literature to know how novel this method is (it seems like using logits to weight all token embeddings is a very natural thing to do; but perhaps the challenge is figuring out the right way to supervise this method in training).

**Questions:**

### Suggestions/comments
- "AIME or IOI problems" acronyms should be defined, or avoided (e.g., "challenging math problems")
- "simplex-weighted compositions": isn't "weighted average" a simpler way of saying the same thing?
- The theoretical perspective on embedding capacity is nice


### Questions
1. Figure 1, $B=8$, why in $t=3$ is there only 1 embedding $e_1$? Why isn't it the mean of all 8 traces? (similar question for $B=2$, $t=3$)
2. For teacher forcing, how is a ground truth prefix $z^*_{<t}$ converted into the continuous tokens $z_{t'}^*$? Where do the weights $\alpha^*_{t'}$ come from? Do you feed the prefix at each step into the current transformer and use its output logits to weight the next token?
3. What model is generating the supervision distribution $\alpha_t^*$? Ah, I see in 3.1 that there are problem-specific approaches to generating $\alpha_t^*$. It would be helpful to specify at the beginning of Section 3 where these $\alpha$ will come from.
4. What is the complexity of MNNS?
5. In proposition 1, what embedding dimension is required for this to work?

---

> ### Author Response · Authors · 2025-11-22
>
> We thank the reviewer for their favorable evaluation of our work and the helpful suggestions. We reply to each of the points raised in the review below. We have also uploaded a revised version of the manuscript highlighting the main changes in red color.
>
> > Q1. "AIME or IOI problems" acronyms should be defined, or avoided (e.g., "challenging math problems")
>
> **Response:** Thank you for flagging this, we edited the introduction by writing the abbreviations in full the first time they appear.
>
> > Q2. "simplex-weighted compositions": isn't "weighted average" a simpler way of saying the same thing?
>
> **Response:** We agree with the reviewer that it is clearer to say "weighted average", but we wanted to make it explicit in the introduction that the weights are obtained from the probability simplex, indicating that coefficients are non-negative and sum to one.
>
> > Q1. Figure 1, $B=8$, why in $t=3$ is there only 1 embedding $\boldsymbol{e}_1$? Why isn't it the mean of all 8 traces? (similar question for $B=2$, $t=3$)
>
> **Response:** Thank you for the question. In the MNNS example there are $m=3$ steps, and at intermediate steps $t=1,2$ we average the embeddings of the states visited by the selected $B$ trajectories. However, as defined in Section 3.1, at **final step** $t=m$ the supervision $\alpha_t^*$ is one-hot with placing all probability mass on the single correct answer. Thus, for $t=3$ we use only $\mathbf{e}_1$, which is the embedding of the minimal non‑negative sum “1”. We updated the Fig. 1 caption to make this clearer according to reviewer's point.
>
> > Q2. For teacher forcing, how is a ground truth prefix $z_{<t}$ converted into the continuous tokens $z_{t'}$? Where do the weights $\alpha^*_{t'}$ come from? Do you feed the prefix at each step into the current transformer and use its output logits to weight the next token?
>
> **Response:** Thank you for raising this question. Under teacher forcing, the ground‑truth prefix is $z_{t'}^\star = E^\top \alpha_{t'}^\star$ where the weights $\alpha_{t'}^\star$ come from our supervision rather than the transformer’s own logits. Conditioned on the previous ground‑truth prefix $z_{t'}^\star$, at each step $t$, we minimize the cross‑entropy between the model’s predicted distribution $\alpha_t$ and the target distribution $\alpha_t^\star$. Concretely, for each task, we construct $\alpha_t^\star$ as follows:
>
> * **MNNS.** We treat sequences of $\{+,-\}$ decisions as trajectories in a search graph. At step $t$, $\alpha_t^\star$ is the normalized distribution over partial sums visited at depth $t$ by the selected trajectories. For the Top‑$B$ filtering, we select $B$ trajectories ranked by the final absolute sum $|s_m|$ and then normalize over the states those $B$ trajectories visit, in the full‑budget ($B = \mathcal{T}$) case we use all trajectories.
>
> * **ProntoQA / ProsQA.** We use reachability in the underlying symbolic graph. At step $t$, $\alpha_t^*$ is the normalized distribution over nodes visited at depth $t$ by the selected trajectories: in the full‑budget case we use all reachable paths. For the Top‑$B$ filtering, we first select $B$ trajectories ranked by the final distance of their endpoint to the correct answer node and then normalize over the states those $B$ trajectories visit, and again we use all trajectories in the full‑budget case.
>
> During inference, however, we do use the model’s own logits $\alpha_t$ to form the continuous token $z_t = E^\top \alpha_t$ and feed it back to the model in autoregressive manner.
>
> > Q3. What model is generating the supervision distribution $\alpha_t^\star$? Ah, I see in 3.1 that there are problem specific approaches to generating $\alpha_t^\star$. It would be helpful to specify at the beginning of Section 3 where these $\alpha_t^\star$ will come from.
>
> **Response:** Thank you for pointing this out. As we stated in the above response, $\alpha_t^\star$ is generated by task‑specific search procedures which corresponds to collecting trajectories, with possibly Top-$B$ trajectory filtering and define $\alpha_t^\star$ as the normalized distribution over the intermediate states visited at step $t$ by these selected trajectories. For MNNS, we enumerate trajectories obtained by visiting one of the two possible next states (adding or subtracting the next integer) at each step, and rank them by the final objective. For ProntoQA and ProsQA, $\alpha_t^\star$ is obtained from the underlying symbolic graph, at step $t$, it is the normalized distribution over all nodes reachable in $t$ hops from the start node.

---

> > ### Author Response · Authors · 2025-11-22
> >
> > > Q4. What is the complexity of MNNS?
> >
> > **Response:** MNNS generalizes classical subset‑sum/partition as stated in Section 3.1.1, and as in subset-sum, one can use dynamic programming in pseudo‑polynomial time $O(mS)$, where $S = \sum_i |d_i|$. The complexity depends on whether input magnitudes are bounded:
> > * **Unbounded inputs.** For $m$ input integers $d_1,\dots,d_m$, we can always define $S = \sum_i |d_i|$ and solve MNNS exactly via a standard dynamic program in time $O(mS)$. However, when the magnitudes $|d_i|$ are arbitrary, $S$ itself can be exponentially large in the bit‑length of the inputs, so this algorithm is only pseudo‑polynomial and MNNS (like subset‑sum) is weakly NP‑hard. In that case, a naive brute‑force search over all sign assignments is an alternative that takes $O(2^m)$ time.
> > * **Bounded inputs.** If the values are bounded, e.g. $|d_i| \le M$ with $M$ polynomial in $m$, then $S = O(mM)$ and the $O(mS)$ algorithm becomes polynomial in $m$.
> >
> > In our experiments we choose a moderate $m$ and bounded magnitudes, so generating CSFT supervision via trajectory enumeration / DP is not expensive.
> >
> > > Q5. In proposition 1, what embedding dimension is required for this to work?
> >
> > **Response:** Thank you for the question. The MNNS construction in App. E.1 uses an embedding split $d = d_e + d_p$ with **$d_e = 2^{n+1}$** to store all $2^k$ partial sums at step $k$ and **$d_p = n+2$** positional coordinates as stated in Assumption 2. So, the sufficient dimension for the full construction over $n$ inputs is $d = 2^{n+1} + (n+2)$. We updated the statement of Proposition 1 to state this explicitly.

---

> > > ### Comment · Reviewer_rbV9 · 2025-11-23
> > >
> > > thanks for the thorough clarifications!

---

### Official Review · Reviewer_sQNo · 2025-10-28

**Soundness:** 3
**Presentation:** 3
**Contribution:** 3
**Rating:** 6
**Confidence:** 3

**Summary:**

The paper presents a technique of chain-of-thought with continuous valued tokens (termed CoT2) for transformer models, and aims to study the effectiveness of this approach over standard CoT and COCONUT baselines. In particular, the authors use the setup with minimum non-negative sum,  ProsQA, and ProntoQA tasks to show how a shallow transformer can utilize CoT2 for parallel search over possible trajectories and avoid error propagation with intermediate step mistakes.

The main idea of CoT2 is to use the softmax outputs (probs) during auto-regressive token generation and consider the probability-weighted sum of embeddings as the continuous-valued token for the subsequent generations. Thus, it captures information from the whole vocabulary instead of committing to a single token. Overall, this work deals with small-scale (2-layer) transformers to study the role of this approach for parallel exploration of trajectories.

**Strengths:**

The work is well presented with small-scale experiments, toy setups, and theoretical results to convey the CoT2 idea (especially the continuous token construction and fine-tuning/RL) and to build intuition about its effectiveness. The authors also discuss potential RL-based extensions for sampling and policy optimization with continuous-valued tokens, which can also inspire future efforts.

**Weaknesses:**

The extension of this approach to practical scenarios is not clear/discussed. Since CSFT requires ground truth $\alpha^*$ for each step, it is unclear how one can compute them for real-world data. There is also no evidence of the benefit of scaling horizontally with more CoT2 tokens in the paper, whereas standard CoT + majority@K (and increasing K) can indeed surpass CoT2.

**Questions:**

1. In Figure 2(a), the majority@K performance of standard CoT with sufficient K surpasses the CoT2 results, and I also noticed that CoT2 does not show any scaling behaviour in the results. Is it because the continuous tokens were always limited to "m" steps?

2. How to decide the "m" value for tasks beyond the toy setup? If a standard CoT approach can solve the task with say A CoT tokens, then can CoT2 solve it with B < A continuous valued tokens? A discussion on how to think about these situations can be helpful.

3. Can a model trained on MNNS with 4 input digits and CoT2 transfer the latent reasoning capability to tasks with 5 inputs? Changing the range of values seems to affect the performance, so how well can CoT2 be transferred to out-of-distribution inputs?

---

> ### Author Response · Authors · 2025-11-22
>
> We thank the reviewer for appreciating the strengths of our work, especially regarding the clarity of our CoT2 formulation, and for the detailed comments. We reply to each of the points raised in the review below and we have also uploaded a revised version of the manuscript highlighting the main changes in red color.
>
> > Weakness 1: The extension of this approach to practical scenarios is not clear/discussed. Since CSFT requires ground truth $\alpha^\star$ for each step, it is unclear how one can compute them for real-world data. There is also no evidence of the benefit of scaling horizontally with more CoT2 tokens in the paper, whereas standard CoT + majority@K (and increasing K) can indeed surpass CoT2.
>
> **Response:** Thank you for raising this point. We would like to first highlight that CSFT does not require an oracle distribution over all intermediate states. Instead, a simple search procedure can be used to collect possible trajectories from the underlying task graphs to obtain $\alpha_t^\star$ by normalizing over the states visited by these trajectories. As a concrete example, in the full budget $B=\mathcal{T}$ case of Fig. 1, the model follows a search procedure over the graph by tracking all reachable nodes at a step. This suggests that tracking multiple reasoning paths in the intermediate steps, without knowing what's the ground-truth intermediate step oracle, can be beneficial.
>
> CSFT is a flexible supervision framework as it specifies how to turn trajectories into token‑level targets, while the source of those trajectories can be adapted to the domain. At a high-level, one can extend supervision to more realistic open-ended task settings by generating trajectories via sampling CoT traces from a stronger teacher using **beam or best‑first search**, and then constructing $\alpha_t^\star$ as the normalized distribution over the states visited by a finite budget $B$ of high‑scoring trajectories which are filtered by a verifier or heuristic. We've added a remark at the end of Section 3.1 discussing the scalability of CSFT to more larger language task settings. On the other hand, CSFT supervision can be skipped when such supervision is hard to obtain, and CoT2 can be trained purely with RL. In particular, our GRPO setup with CoT2‑MTS treats the continuous token as the action and uses only a sparse final‑answer reward, so continuous tokens can be learned even without intermediate $\alpha_t^\star$.
>
> Regarding the second part of the comment, it's expected that Pass@$K$ with standard CoT, which uses **$K$ independent** rollouts, will eventually surpass a **single** greedy CoT2 rollout in Fig. 2a. We direct the reviewer to our response to Q1 below, where we show that applying Pass@K with CoT2 outperforms discrete CoT with Pass@$K$ at each $K$.
>
> > Q1. In Figure 2(a), the majority@K performance of standard CoT with sufficient K surpasses the CoT2 results, and I also noticed that CoT2 does not show any scaling behaviour in the results. Is it because the continuous tokens were always limited to "m" steps?
>
> **Response:** We thank the reviewer for the thoughtful question. As we also explain in the answer to Q2 below, in our experiments, the number of reasoning steps $m$ is fixed across methods by the task structure: for MNNS, each input digit is processed once, so $m$ equals the number of input digits; for the ProsQA/ProntoQA, it corresponds to the predetermined number of hops. In Fig. 2a, we vary the *number of independent rollouts* generated from discrete CoT, and therefore we observe a scaling behavior by aggregating them with Pass@K. For CoT2 model, the lack of scaling behavior is because it uses **single‑shot greedy decoding with temperature (T=0)**.
>
> To further address the reviewer’s point, we also ran **Pass@K** on top of the CoT2 model by sampling at the final step, using temperatures varying from $0$ to $1$ instead of the fixed $T=0$ setting used in Fig. 2a. In this experiment, CoT2 shows clear scaling with $K$, and for each $K$ the CoT2+Pass@K curve consistently outperforms the corresponding discrete CoT+Pass@K. These results can be accessed from the [link](https://anonymous.4open.science/r/Continuous-Chain-of-Thought-Parallel-Exploration-and-Reasoning-through-a-Theoretical-Lens-01D6/Pass_k_Embed_24_Layers_1_Heads_1.pdf) here, and we have also added this plot to the Appendix D.1 by revising the manuscript in response to the reviewer's comment.

---

> ### Author Response · Authors · 2025-11-22
>
> > Q2. How to decide the "m" value for tasks beyond the toy setup? If a standard CoT approach can solve the task with say A CoT tokens, then can CoT2 solve it with B < A continuous valued tokens? A discussion on how to think about these situations can be helpful.
>
> **Response:** We thank the reviewer for the insightful question. Although our methods CSFT or GRPO-MTS admit different choices of the reasoning length $m$, as noted in our response to Q1, in our experiments $m$ is inherited from the mechanistic structure of the task. Our main focus in this work is to understand **how much parallel reasoning CoT2 can perform** per step under a fixed reasoning length, therefore, shortening the reasoning length $m$ is an interesting but orthogonal direction to the contribution of this paper.
>
> On the other hand, models can also generate long CoT traces, so a discrete CoT strategy with Pass@$K$ can effectively be viewed as a single long CoT trajectory with reasoning length $K \cdot m$. On MNNS task in Fig. 2a, a **single** CoT2 trajectory matches the accuracy of discrete CoT with multiple ($\sim 10$) independent samples, while using only $m$ continuous tokens and packing multiple trajectories into each step. In that sense, for a fixed $m$, CoT2 achieves a given accuracy with a **much shorter effective reasoning length** than standard CoT that relies on many rollouts. This suggests that, in more general settings, a practical recipe for selecting $m$ should increase with the desired target accuracy but can decrease as the embedding capacity $d$ grows, since a larger $d$ allows more parallelism to be packed into each decoding step of CoT2 according to our information‑packing bound in App. E.
>
>
> > Q3. Can a model trained on MNNS with 4 input digits and CoT2 transfer the latent reasoning capability to tasks with 5 inputs? Changing the range of values seems to affect the performance, so how well can CoT2 be transferred to out-of-distribution inputs?
>
> **Response:** We thank the reviewer for the question. We believe that out‑of‑distribution generalization this is a broader issue related to  is not specific to CoT2. If a discrete CoT model generalizes in a given setting, then CoT2 should **also generalize** with a fixed budget $B$ since it simply tracks them $B$ discrete CoT traces in parallel within each step, provided the embedding capacity $d$ is sufficient for representation.
>
> Regarding the length generalization, because our models are standard transformer decoders rather than state‑space models (e.g., Mamba, H3), it is not the right expectation that a model trained only on 4‑digit MNNS will generalize well to 5‑digit MNNS. We therefore believe that it's hard to claim strong zero‑shot transfer in these specific tasks that are studied.

---

### Official Review · Reviewer_JDh5 · 2025-10-31

**Soundness:** 3
**Presentation:** 2
**Contribution:** 2
**Rating:** 4
**Confidence:** 3

**Summary:**

This paper proposes an interesting framework termed _Continuous Chain of Thought_ (CoT2). Inspired by a previous work titled chain of continuous thought (COCONUT), this work trains transformer models to reason in a continuous token space rather than via discrete linguistic steps. Each reasoning token is represented as a convex combination of vocabulary embeddings, allowing multiple reasoning trajectories to coexist in superposition. The model is trained from scratch using a Continuous Supervised Fine-Tuning (CSFT) objective that aligns predicted token distributions with soft targets derived from the top-BBB teacher trajectories, followed by a Generalized Reinforcement Policy Optimization (GRPO) phase to reinforce correct reasoning paths. Experiments on synthetic reasoning benchmarks such as MNNS and ProsQA show that this approach achieves higher task accuracy compared with standard discrete CoT baselines. In addition, the authors revealed a link between embedding dimensionality and the number of representable reasoning branches via theoretical analysis.

**Strengths:**

- The framework offers a theoretically principled formulation of parallel reasoning within neural sequence models, on a scale that is testable with single GPUs.
- The CSFT objective is mathematically clean, converting multi-trajectory supervision into a single convex target, and the accompanying proofs clarify the representational capacity of continuous token mixtures.
- The paper establishes a precise connection between continuous probabilistic reasoning and information-theoretic efficiency, supported by consistent empirical trends in accuracy and entropy reduction.

**Weaknesses:**

- The study’s experimental scope and scale are limited. All models are toy-sized and trained from scratch with pruned vocabularies on synthetic tasks and symbolized formulations that departure significantly from natural languages, making it unclear whether the proposed principles extend to large, pretrained language models operating over natural text.
- For reasons explained above, I am not convinced that the comparisons against conventional CoT and COCONUT are fair. The uncontrolled and unfair nature of the benchmarking obscures the contribution and significance of the results.
- Moreover, the self-designation “CoT2” implies direct lineage from the original chain of thoughts. I find this misleading since the proposed method differs fundamentally from both discrete CoT prompting and latent-space reasoning approaches like COCONUT.

**Questions:**

- Can the proposed continuous-token formalism scale to full-vocabulary pretrained LMs, or is it confined to small synthetic domains? No experiments needed if the authors are pressed for time. Simple speculation suffices.
- How sensitive are results to the trajectory budget B and to the number of reasoning steps? Can one set these hyper-parameters automatically?

---

> ### Author Response · Authors · 2025-11-22
>
> We thank the reviewer for the thoughtful comments and for acknowledging the potential impact of the work. Below we respond to each of the points raised. We have also uploaded a revised version of the manuscript highlighting the main changes in red color.
>
> > Weakness 1: The study’s experimental scope and scale are limited. All models are toy-sized and trained from scratch with pruned vocabularies on synthetic tasks and symbolized formulations that departure significantly from natural languages, making it unclear whether the proposed principles extend to large, pretrained language models operating over natural text.
>
> **Response:** We thank the reviewer for this comment. We believe that our work makes fundamental contributions to this research area by formulating a continuous decoding mechanism on the embedding space, developing a generic supervision framework with continuous tokens and providing a theoretical link to parallelism, which offers useful insights for scaling to larger language models.
>
> We would also like to highlight that CoT2’s continuous tokens are simplex‑weighted combinations of vocabulary embeddings, and our CSFT supervision **explicitly targets parallel reasoning** by tracking distributions over multiple trajectories at each step with a controllable trajectory budget $B$. This makes our framework particularly **well suited to problems requiring search** in the underlying task graph, such as ProsQA, ProntoQA and MNNS, which is why our main experimental focus is on these tasks rather than on natural‑language benchmarks. That said, we agree with the reviewer that it'd be valuable to apply CoT2 to more complex settings, and we plan to explore applying GRPO + CoT2‑MTS starting from an SFT baseline on GSM8K during this rebuttal period, if time permits.
>
>
> > Weakness 2: For reasons explained above, I am not convinced that the comparisons against conventional CoT and COCONUT are fair. The uncontrolled and unfair nature of the benchmarking obscures the contribution and significance of the results.
>
> **Response:** We thank the reviewer for their feedback. While we understand the concern that we did not evaluate on more complex reasoning benchmarks, we would like to clarify that the tasks ProntoQA, ProsQA, and MNNS we evaluated our method on are proper logical-reasoning tasks, and the reported results **correctly reflect algorithmic differences** between baselines. We respectfully disagree with the reviewer's view that this makes our evaluations unfair as **all baselines share the same architecture, are trained from scratch on the same datasets, and use matched training budgets** (same number of epochs, with COCONUT’s curriculum stages sharing that budget as described in App. C.3) and matched inference settings.
>
> > Weakness 3: Moreover, the self-designation “CoT2” implies direct lineage from the original chain of thoughts. I find this misleading since the proposed method differs fundamentally from both discrete CoT prompting and latent-space reasoning approaches like COCONUT.
>
> **Response:** Thank you for the question. We use “CoT2” as shorthand for Chain of Thought with Continuous Tokens to make it explicit that we still operate in a step‑by‑step reasoning regime but **exploit the probability simplex** to obtain a richer representation by **superposing different reasoning trajectories** in a single token.
>
> At the same time, we believe that CoT2 is **algorithmically closely related** to COCONUT rather than being a completely different paradigm. To be more concrete, if $h_t$ denotes the final‑layer hidden state at step $t$, COCONUT feeds $h_t$ itself to the next step as a latent token, whereas CoT2 first maps $h_t$ to logits $\ell_t = W_{\text{out}} h_t$, converts them to a distribution $\alpha_t = \mathrm{softmax}(\ell_t)$, and then projects back to the embedding space via $z_t = E^\top \alpha_t$, which is fed forward as the next “continuous token”.  When $\alpha_t$ is one‑hot (corresponding to $B=1$), this reduces exactly to the **discrete CoT**, and when we drop the $E^\top$ projection and use $h_t$ directly, we recover a **COCONUT‑style** latent token. Thus, we believe that CoT2 is **not a fundamentally different framework** from these two approaches, but rather a specific way of keeping continuous tokens in the vocabulary embedding space so model can emulate parallel reasoning over different trajectories.

---

> > ### Author Response · Authors · 2025-11-22
> >
> > > Q1. Can the proposed continuous-token formalism scale to full-vocabulary pretrained LMs, or is it confined to small synthetic domains? No experiments needed if the authors are pressed for time. Simple speculation suffices.
> >
> > **Response:** We thank the reviewer for this question. Architecturally, the base CoT2 decoding mechanism $z_t = E^\top \alpha_t$ applies directly to larger pretrained models and this one extra matrix–vector multiply adds only a small overhead to the inference cost (please see our response to W3 of reviewer EUvY). Our **CSFT is generic** as it specifies how to turn a set of trajectories into token‑level targets, while leaving the choice of trajectories to the task at hand. Together with that, our RL sampling algorithms (MTS and Dirichlet sampling) are **directly compatible** with pretrained language models.
> >
> > Since CSFT is designed to work with any set of trajectories, the main effort lies in scaling to larger vocabularies and more open-ended tasks is constructing the supervision $\alpha_t^\star$ by collecting multiple trajectories and normalizing over the visited states at each step. In this work, we build $\alpha_t^\star$ by collecting trajectories from the graph structure of the MNNS, ProntoQA, ProsQA tasks. A similar idea can be extended by constructing $\alpha_t^\star$ from multiple CoT trajectories sampled from a stronger teacher using **beam or best-first search** and aggregated into a token‑level distribution, which can then be filtered by a verifier or heuristic and normalized over the selected paths.
> >
> > In settings where such intermediate supervision is hard to obtain, CSFT can be skipped altogether and CoT2 can be trained via GRPO in the **continuous action space** with CoT2‑MTS sampling and using only a sparse final‑answer reward. For this, one may need a curriculum learning approach to gradually replace a set of discrete tokens by continuous tokens. Regarding the reviewer's comment, we have added a brief discussion at the end of Section 3.1 on the scalability of CSFT to larger‑scale language settings.
> >
> > > Q2. How sensitive are results to the trajectory budget B and to the number of reasoning steps? Can one set these hyper-parameters automatically?
> >
> > **Response:** Thank you for the insightful question. As we describe in Sections 3.1.1 and 3.1.2, the number of reasoning steps $m$ is fixed by the task structure in our experiments, for MNNS it equals the number of input digits, and for ProntoQA/ProsQA it corresponds to the pre-determined number of hops (i.e., graph diameter). Our main goal in this work is to understand how reasoning can be made more efficient with CoT2 *under a fixed reasoning length*. Therefore, reducing the number of reasoning steps $m$ is a valuable but orthogonal direction to our contribution.
> >
> > On the other hand, how we should select $B$ depends on the embedding dimension $d$ (and not directly to the number of steps). As we stated in 3.2, we observe in Fig. 2c and Fig. 3 that the optimal $B$ increases with embedding dimension $d$. Accordingly, a rule of thumb could be selecting $B \log(v/B) \propto d $ based on our information-packing bound in App. E, which keeps the **“amount of packed parallelism”** roughly matched to the available embedding capacity.
> >
> > As also noted in our response to the Q2 of Reviewer sQNo, intuitively, CoT2 with parallelism budget $B$ can be viewed as a single discrete long CoT rollout of length $B \cdot m$ with **B** repeated internal trials. In practice, once $B$ is determined based on the available embedding capacity, one can pick the smallest $m$ that reaches a target Pass@$K$ accuracy using length $m$ rollouts with $K\approx B$.

---

> ### Comment · Reviewer_JDh5 · 2025-11-27
>
> I thank the authors for the comprehensive reply. My skepticisms are satisfactorily addressed and I have updated the score accordingly.

---

### Official Review · Reviewer_oWRB · 2025-10-31

**Soundness:** 3
**Presentation:** 3
**Contribution:** 3
**Rating:** 6
**Confidence:** 3

**Summary:**

The paper proposes CoT2, a new method for continuous chain of thought, which constructs continuous tokens as a linear superposition of tokens. Such models can be trained via SFT on a combination of top-K teacher traces. It is shown theoretically that a one-layer transformer construction can solve a version of subset-sum using CoT2 by tracking multiple computations in parallel. Moreover, it is shown that decoding with $N$ chains of CoT2 by averaging top $K$ sampled tokens, is upper bounded by the expected error of $NK$ ordinary CoT chains, showing the statistical benefits of parallelism. Finally, a GRPO objective is proposed for continuous CoT.

**Strengths:**

* The proposed CoT2 method is novel and simple to implement on top of existing models.
* Since each continuous token is a superposition of discrete tokens, the CoT should be more interpretable compared to methods which simply append the hidden representations.
* Theoretical analysis supports the intuitive benefits of tracking multiple traces. In particular, the increased information content is reflected in improved sample complexity.
* It is demonstrated that GRPO with MTS can adapt models to continuous rollouts and improve performance on synthetic tasks.

**Weaknesses:**

* While each continuous token should carry $K$ times more information than a discrete token with MTS as demonstrated by the sample complexity results, we also need to sample $K$ times more tokens when decoding. What is the tradeoff in terms of actual runtime (during SFT, decoding, RL, etc)?
* While a transformer construction solving MNNS with CoT2 is provided in Proposition 2, there is no lower bound for ordinary CoT, os it is unclear whether this constitutes an actual improvement over discrete CoT.
* The GRPO policy ratio proposed in Eq.(4) is not an unbiased estimate via policy gradient theorem.
* Since GRPO with MTS can be used to adapt existing discrete models to output continuous CoT, I would expect to see performance on larger models than GPT-2 and more challenging reasoning benchmarks. Can the authors add any larger-scale experiments?

**Questions:**

* As mentioned by the authors, the CSFT training scheme can be seen as token level distillation. Can the authors give a comparison with related methods in distillation?
* See also the questions in Weaknesses.

---

> ### Author Response · Authors · 2025-11-22
>
> We appreciate the reviewer’s positive assessment of our work and their detailed feedback. We address all of the points below.
>
> > Weakness 1: While each continuous token should carry $K$ times more information than a discrete token with MTS as demonstrated by the sample complexity results, we also need to sample $K$ times more tokens when decoding. What is the tradeoff in terms of actual runtime (during SFT, decoding, RL, etc)?
>
> **Response:** We thank the reviewer for the thoughtful question. Proposition 3 shows that MTS with parallelism $K$ is $K$ times more sample-efficient than discrete CoT. However, for the actual runtime, one forward pass of MTS is far **cheaper** than $K$ forward passes of discrete CoT. This is because we make one forward pass to obtain $\alpha_t$, sample $K$ tokens from it, average their embeddings, and proceed. The only extra cost here is doing $K$ embedding lookups instead of a single lookup and an averaging operation, both of which are much cheaper than the cost of even one additional forward pass through the whole transformer stack. By contrast, discrete CoT requires $K$ independent forward passes to reach similar accuracy, which is **substantially more costly** than MTS with parallelism $K$.
>
> > Weakness 2: While a transformer construction solving MNNS with CoT2 is provided in Proposition 2, there is no lower bound for ordinary CoT, so it is unclear whether this constitutes an actual improvement over discrete CoT.
>
> **Response:** We appreciate the valuable comment. The motivation behind Proposition 1 is to theoretically formalize an empirical phenomenon we observed. In our experiments (Fig. 7 in Appendix D), a 1‑layer CoT2 model can solve the MNNS task with near‑perfect accuracy once it is given enough embedding capacity $d$ to superpose several partial‑sum representations, whereas a 1‑layer discrete CoT model with the **same** embedding dimension $d$ still struggles. The main bottleneck for discrete CoT in this setting is that it can only represent a single state at each step, making it information‑inefficient compared to CoT2 and forcing it to rely on multiple rollouts to explore different trajectories.
>
> We do not claim a formal impossibility lower bound for discrete CoT on MNNS in this work, rather, our experiments show that discrete CoT does not reach perfect accuracy under matched conditions. However, we agree with the reviewer that establishing such a lower bound under matching embedding dimension constraints is an interesting direction to explore.
>
> > Weakness 3: The GRPO policy ratio proposed in Eq.(4) is not an unbiased estimate via policy gradient theorem.
>
> **Response:** We thank the reviewer for bringing this. The reviewer is correct that the policy ratio in Eq. (4) is not an unbiased estimator of the policy gradient; however, we would like to note that GRPO/PPO-style methods are also biased surrogates of the policy gradient because of the clipping operator and the KL penalty term. In our MTS approach at step $t$, the joint action is the $K$-tuple $(i_1,\ldots,i_K)$ with probability $\prod_{r=1}^{K}\alpha_{t,i_r}$. So, the exact policy ratio would be $\prod_r \frac{\alpha_{t,i_r}}{\alpha_{t,i_r}^{\mathrm{old}}}$. Similar to GRPO/PPO‑style clipped objectives, we do not optimize this exact joint ratio $\prod_r \frac{\alpha_{t,i_r}}{\alpha_{t,i_r}^{\mathrm{old}}}$ directly, because products of $K$ probabilities can be numerically unstable. Instead, we use its geometric mean $r_t=\exp \big(\tfrac{1}{K}\sum_r \log \alpha_{t,i_r} - \tfrac{1}{K}\sum_r \log \alpha_{t,i_r}^{\mathrm{old}}\big)$ in Eq. (4) to normalize scale across $K$ and stabilize updates by avoiding numerical overflows. When per‑token ratios are close to 1 (which is the regime enforced by the clipping), we can approximate $r_t$ by a first‑order Taylor expansion around $r_t = 1$ so that $r_t \approx 1 + \tfrac{1}{K}\sum_r \big(\log \alpha_{t,i_r}-\log \alpha_{t,i_r}^{\mathrm{old}}\big).$ Plugging this approximation into the GRPO term $r_t \hat A_t$ gives a gradient proportional to $\tfrac{1}{K}\sum_r \nabla \log \alpha_{t,i_r} \hat A_t$, which is exactly the joint‑action policy‑gradient up to the constant factor $1/K$, and this constant does not change the optimizer as it's absorbed by the learning‑rate. In other words, Eq. (4) defines a proxy objective whose local gradient well-approximates the true joint‑action objective.

---

> ### Author Response · Authors · 2025-11-22
>
> > Weakness 4: Since GRPO with MTS can be used to adapt existing discrete models to output continuous CoT, I would expect to see performance on larger models than GPT-2 and more challenging reasoning benchmarks. Can the authors add any larger-scale experiments?
>
> **Response:** We thank the reviewer for this suggestion. Our CSFT supervision is implicitly well suited for enabling parallel reasoning, as it constructs distributions over the states visited by a selected trajectories at each step. For this reason, our main experimental focus in this paper is on search‑heavy tasks such as MNNS, ProntoQA, and ProsQA where these distributions can be obtained from the underlying graph. On the other hand, we agree with the reviewer that it would be interesting to explore CoT2 on math reasoning tasks, and during the rebuttal period we plan to try explore GRPO training with CoT2‑MTS on GSM8K from an SFT baseline if time permits.
>
>
> > Q1. As mentioned by the authors, the CSFT training scheme can be seen as token level distillation. Can the authors give a comparison with related methods in distillation?
>
> **Response:** We thank the reviewer for the insightful question. CSFT can indeed be viewed as a form of token‑level distillation since both approaches define their loss functions using divergence of token distributions, but it differs from standard distillation in both the **source of supervision** and the **aim of the distillation**. Classical CoT distillation methods use a large teacher LLM to generate natural‑language trajectory and then train a smaller student to imitate those step‑by‑step logits by minimizing a divergence $D\big(\mathrm{LM}\_{\\mathrm{teacher}}(\cdot \mid z\_{<t}) || \mathrm{LM}_{\mathrm{student}} (\cdot \mid z\_{<t}) \big)$[1, 2]. In contrast, CSFT has a more flexible source of supervision. Besides being able to use CoT trajectories produced by a teacher LLM, it can also rely on supervision $\alpha_t^\star$ constructed from the trajectories in the underlying task graph, possibly with filtering to select a subset of trajectories, and then a divergence $D(\alpha_t^\star || \alpha_t)$ to distill these state distributions into the vocabulary simplex. Thus, the main difference lies in CSFT **tracking multiple trajectories in token space** with a distribution over vocabulary, while standard distillation focuses on **fully matching the teacher’s conditional distributions** over a single CoT trajectory.
>
> [1] Peifeng Wang, Zhengyang Wang, Zheng Li, Yifan Gao, Bing Yin, and Xiang Ren. SCOTT: Self‑Consistent Chain‑of‑Thought Distillation. ACL 2023.
>
> [2] Cheng‑Yu Hsieh, Chun‑Liang Li, Chih‑Kuan Yeh, Hootan Nakhost, Yasuhisa Fujii, Alexander Ratner, Ranjay Krishna, Chen‑Yu Lee, and Tomas Pfister. Distilling Step‑by‑Step! Outperforming Larger Language Models with Less Training Data and Smaller Model Sizes. Findings of ACL 2023.

---

### Author Response · Authors · 2025-11-22
**Main Response to All Reviewers**

We thank the reviewers for their thoughtful and constructive feedback. We are encouraged by their recognition of the potential impact of our work, which we see as laying a foundation for continuous‑token reasoning: (i) we **formalize decoding in the vocabulary embedding space** ($z_t = E^\top \alpha_t$), (ii) introduce a **generic, budget‑controlled continuous supervision** framework CSFT that tracks parallel trajectories, and (iii) develop **RL sampling methods in a continuous action space** (Multi‑token and Dirichlet sampling) together with a stable GRPO objective for CoT2, and (iv) providing theory that **quantifies CoT2’s parallelism and efficiency**, and an **information‑packing bound** linking **embedding dimension $d$** to the **parallelism budget $B$**. In response to the reviewers' suggestions, we have uploaded a revised manuscript with changes shown in red. The revisions can be summarized as follows:

* We updated Fig. 2a and the accompanying discussion to now apply Pass@k to both CoT2 and discrete CoT, where CoT2 consistently outperforms at every $k$, and its single‑shot performance matches the multi‑shot discrete CoT results. To make the comparison clearer and demonstrate CoT2’s scaling behavior, we replaced greedy (temperature=0) final‑step decoding with sampling at temperatures $T\in[0,1]$.
* We revised the caption of Fig. 1 and the statement of Proposition 1 for clarity.
* We added a remark at the end of Section 3.1 discussing the scalability of CSFT to larger language‑model settings.
* We added another remark at the end of Section 2 that clarifies the scalability of the base CoT2 decoding by discussing the inference‑time cost of the additional matrix–vector multiplication required to obtain the continuous token $z_t = E^\top\alpha_t$.

Together with these, according to raised points to evaluate our method on reasoning benchmarks beyond logical multi-hop reasoning, we ran **Qwen3-0.6B** on **GSM8K** using our **CoT2-MTS** continuous-token sampling procedure. We implemented the continuous rollout/training code from scratch, and because we couldn't leverage frameworks such as vLLM to speed up generation, we conducted our experiments on modest model sizes and decoding budgets. For CoT2-MTS, we used a **curriculum learning** approach where continuous tokens are introduced from **50%** of training, ramped to full by **80%**, and used thereafter. The table below reports results for different maximum response lengths (160, 128, 96, 80 tokens) and we set the number of superposed tokens $K=4$ for all CoT2-MTS settings. We use 16 continuous tokens for the 96/80 settings and 32 continuous tokens for the 128/160 settings, for example, in the 160-token configuration the effective budget grows from 128 discrete tokens at the start of training to 128 discrete + 32 continuous tokens (i.e., 128->160) as the curriculum completes.

| Max Resp. Len | Initial Acc (%) | Discrete CoT Acc (%) | CoT2-MTS Acc (%) | Avg. Len (Disc CoT) | Avg. Len (CoT2-MTS) |
| :-----------: | :----------: | :------------------: | :--------------: | :-----------------: | :-----------------: |
|           160 |         34.6 |                 37.0 |             **41.3** |              **97.3** |             114.8 |
|           128 |         27.6 |                 34.3 |             **36.2** |             104.8 |              **96.6** |
|            96 |         17.2 |                 **27.2** |             27.1 |              84.0 |              **83.7** |
|            80 |          9.5 |                 21.5 |             **27.0** |              73.9 |              **67.0** |

Across GSM8K with Qwen3‑0.6B, CoT2‑MTS improves over discrete CoT in **three out of four** maximum response length settings and matches it in the remaining one, while also shortening the average response length. Overall, these preliminary results indicate that performing exploration in the continuous action space via CoT2-MTS proves to be a promising direction. We provide a more detailed description of these experiments in Appendix D.2.

---

### Author Response · Authors · 2025-12-03
**Rebuttal Summary for Area Chair**

Dear Area Chair, below we briefly summarize the main updates made during the rebuttal period:
* **Points of agreement.** Reviewers broadly agreed that our approach is *natural and intuitive*, the presentation is *clear*, and the paper and the topic are *high‑impact*. They also highlighted that the paper offers a blend of theory and experiments, contrasting sample-complexities and packing-bounds for different decoding methods, and introducing the novel methods of supervising continuous tokens with CSFT and sampling in the continuous action space with CoT2-MTS.
* **Score changes.** Among the two reviews that initially assigned a score of **4**, **Reviewer JDh5** raised their score to **6** after our clarifications by stating that their **skepticism was addressed satisfactorily**. Also, Reviewer EUvY stated that our responses **answer most of their concerns**. Following their response, we further ran preliminary GSM8K experiments with CoT2-MTS to address their point on additional benchmarks.
* **Key clarifications.** We clarified (i) the **scalability** of base CoT2 decoding, CSFT supervision, and CoT2‑MTS RL to larger LMs, (ii) that CSFT does not require oracle intermediate labels and how supervision is built from task‑specific / teacher trajectories, (iii) the role of the MNNS task and its link to our theory, (iv) the justification of the geometric-mean policy ratio used in GRPO for CoT2-MTS.
* **Additional results and revisions.** We highlighted our changes to the paper during rebuttal in **red** color. Responding to EUvY’s feedback on testing our methods on different benchmarks, we ran preliminary **GSM8K** experiments with **Qwen3‑0.6B** using **CoT2‑MTS with curriculum learning**, where the results are summarized in the table below. Also, in response to Reviewer sQNo, we updated Fig 2.a by also applying Pass@k on top of CoT2, to contrast its scaling behavior against that of discrete CoT.

In the next main response, we summarized our revisions during rebuttal.

---

### Meta-Review · Area_Chair_fLcM · 2026-01-10

**Summary:**

The paper introduces *Continuous Chain of Thought (CoT2)*, a framework that moves beyond discrete token sampling to perform reasoning with continuously-valued tokens. The authors provide a theoretical lens through which they establish how continuous tokens allow a model to track multiple reasoning traces in parallel, thereby improving inference efficiency. They prove that a one-layer transformer using CoT2 can solve the combinatorial subset sum problem. Methodologically, the work introduces a novel supervision strategy based on empirical token distributions and a multi-token sampling (MTS) strategy that enables policy optimization for continuous reasoning traces.

This paper provides an useful and novel framework that potentially improves the way of implementing chain of thought. Especially, following points are valuable:
- Novel Framework with Intuitive Appeal: The proposed CoT2 framework is both theoretically motivated and intuitively more flexible than standard discrete Chain of Thought. By allowing the model to operate in a continuous latent space, it facilitates implicit parallel search, which is a significant conceptual advancement in LLM reasoning.
- Methodological Novelty: The introduction of policy optimization for continuous tokens is a novel contribution that unlocks new ways to refine reasoning capabilities beyond initial supervision.


Although the reviewers raised several concerns, the authors proactively provided replies to them during the rebuttal phase. Initial concerns regarding the scale of experiments were effectively addressed by providing additional results using Qwen2.5-0.5B on the GSM8K dataset with the MTS strategy. These new experiments demonstrated that the benefits of CoT2 scale to more modern, practical architectures beyond toy models.
Another concern on the additional computational cost induced by $K$ parallelization was also properly addressed in their rebuttals.

In summary, the paper presents a novel approach to sequence modeling and reasoning. The theoretical proofs are solid, and the empirical concerns raised by reviewers were convincingly addressed, especially, through the inclusion of larger-scale model experiments during the rebuttal. The work is highly relevant to the ICLR community and provides an interesting insight on LLM developments. Thus, I recommend acceptance.

**Reviewer Concerns:**

I did not find any concern on reviews.

**Reviewer Scores:**

The scores of Reviewer JDh5 and EUvY would have been raised from 4 to 6 for both reviewers. Acually, Reviewer JDh5 is mentioning that they raised their score.

---

### Decision · Program_Chairs · 2026-01-26

Accept (Poster)